# The RealHumanEval: Evaluating Large Language Models' Abilities to Support Programmers

**Hussein Mozannar\***                                    *hmozannar@microsoft.com*
*Microsoft Research*

**Valerie Chen\***                                        *vchen2@andrew.cmu.edu*
*Carnegie Mellon University*

**Mohammed Alsobay**                                      *mosobay@mit.edu*
*Massachusetts Institute of Technology*

**Subhro Das**                                            *subhro.Das@ibm.com*
*MIT-IBM Watson AI Lab*
*IBM Research*

**Sebastian Zhao**                                        *sebbyzhao@berkeley.edu*
*University of California, Berkeley*

**Dennis Wei**                                            *dwei@us.ibm.com*
*MIT-IBM Watson AI Lab*
*IBM Research*

**Manish Nagireddy**                                      *manish.nagireddy@ibm.com*
*MIT-IBM Watson AI Lab*
*IBM Research*

**Prasanna Sattigeri**                                    *psattig@us.ibm.com*
*MIT-IBM Watson AI Lab*
*IBM Research*

**Ameet Talwalkar**                                       *atalwalkar@gmail.com*
*Carnegie Mellon University*

**David Sontag**                                          *dsontag@csail.mit.edu*
*Massachusetts Institute of Technology*

*\*: Equal contribution.*

**Reviewed on OpenReview:** *https://openreview.net/forum?id=M7SO74I9mo*

## Abstract

Evaluation of large language models for code has primarily relied on static benchmarks, including HumanEval (Chen et al., 2021), or more recently using human preferences of LLM responses. As LLMs are increasingly used as programmer assistants, we study whether gains on existing benchmarks or more preferred LLM responses translate to programmer productivity when coding with LLMs, including time spent coding. We introduce `RealHumanEval`, a web interface to measure the ability of LLMs to assist programmers, through either autocomplete or chat support. We conducted a user study (N=243) using `RealHumanEval` in which users interacted with seven LLMs of varying base model performance. Despite static benchmarks not incorporating humans-in-the-loop, we find that improvements in benchmark performance lead to increased programmer productivity; however gaps in benchmark ver-

sus human performance are not proportional—a trend that holds across both forms of LLM support. In contrast, we find that programmer preferences do not correlate with their actual performance, motivating the need for better proxy signals. We open-source `RealHumanEval` to enable human-centric evaluation of new models and the study data to facilitate efforts to improve code models.

## 1 Introduction

Coding benchmarks such as HumanEval (Chen et al., 2021) and MBPP (Austin et al., 2021) play a key role in evaluating the capabilities of large language models (LLMs) as programming becomes a valuable application through products such as GitHub Copilot (Github, 2022) and ChatGPT (OpenAI, 2022). These benchmarks quantify LLM abilities by measuring how well a model can complete entire coding tasks. As LLMs are increasingly adopted as programmer assistants—providing chat responses or autocomplete suggestions, rather than full code generations—prior works have argued for bringing humans-in-the-loop to evaluate LLMs (Lee et al., 2023; Chiang et al., 2024). A predominant human-centric approach collects human preference judgments of intermediate LLM outputs, whether between pairs of LLM responses (e.g., Chatbot Arena (Chiang et al., 2024)) or, for coding in particular, using programmer acceptance rates of LLM suggestions (e.g., in products such as Github Copilot (Bird et al., 2022)). However, such evaluation may not capture the LLM's downstream impact on programmer productivity.

Evaluating the utility of LLMs on downstream productivity requires conducting user studies where programmers code with LLM assistance. While a set of small-scale user studies have been conducted to primarily build a qualitative understanding of how programmers use LLM assistance, they are typically restricted to evaluations on one model and one form of LLM support, primarily relying on commercial tools like Github Copilot or ChatGPT (Barke et al., 2023; Mozannar et al., 2024; Vaithilingam et al., 2022; Ross et al., 2023; Liang et al., 2023; Peng et al., 2023). To enable evaluations of a broader set of LLMs and lower the barrier to conducting these studies, we introduce an online evaluation platform, `RealHumanEval`[1] (Figure 1). The platform consists of a code editor where programmers can solve coding tasks with two common forms of LLM assistance: programmers can either ask questions to the LLM through a chat window or receive code completion suggestions through an autocomplete system inside the editor. The interface also supports executing and testing code and logging telemetry which can be used to compute productivity metrics, which we operationalize as the time to complete a task or number of tasks completed, and preference metrics, including average acceptance rates of suggestions and the likelihood of copying code from chat responses.

Using `RealHumanEval`, we conduct a user study with 243 participants to understand the effect of a model's benchmark performance and the form of LLM assistance on time to complete a task and number of tasks completed. Each participant was assigned to one of seven conditions: a control condition with no LLM support, three conditions with autocomplete support from either `CodeLlama-7b` (Rozière et al., 2023), `CodeLlama-34b` (Rozière et al., 2023), or `GPT-3.5-turbo-instruct`(Brown et al., 2020), and finally three conditions where the editor is equipped with a chat window powered by the chat variants of the previous models in addition to `GPT-4o` (OpenAI, 2022). We deliberately select model families with increasingly higher benchmark performance and consider model pairs within each family with similar benchmark performance to understand the effect of autocomplete versus chat assistance. Through the study, we collected a dataset of interactions on 888 coding total tasks, where 5204 autocomplete suggestions were shown and 1055 chat messages were sent.

Overall, we find that improving a model's base performance on existing coding benchmarks leads to improvements in the time spent completing tasks. These trends were present across both chat and autocomplete interactions, validating the potential "generalizability" of benchmarks to more realistic contexts. However, we observe that gaps in benchmark versus human performance are not necessarily proportional, suggesting that further gains in benchmark performance do not necessarily translate into equivalent gains in human task completion. We also investigated whether human preference metrics, such as the average acceptance rate

---

[1] The choice of naming our platform `RealHumanEval` is to imply that the evaluation is done with the help of real humans. It does not imply that it is a real evaluation while other benchmarks are not real, nor does it imply that this is the real version of the HumanEval benchmark.

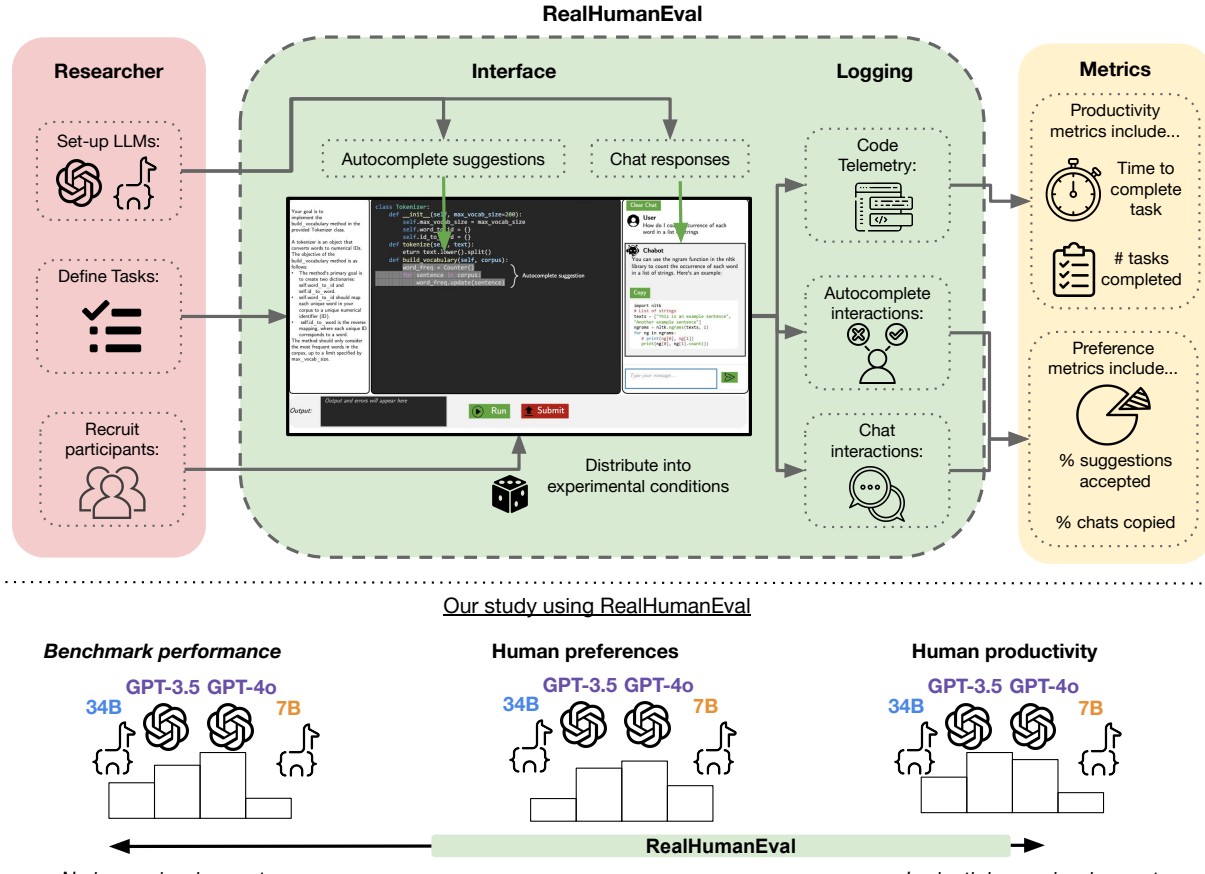

Figure 1: We introduce `RealHumanEval`, an end-to-end online evaluation platform of LLM-assisted coding through autocomplete suggestions and chat support. The goal of `RealHumanEval` is to facilitate human-centric evaluation of code-generating LLMs, simplifying the workflow for researchers to conduct user studies to measure the effect of LLM assistance on downstream human productivity and preferences. We selected 4 families of LLMs of varying sizes (`GPT-4o`, `GPT-3.5`, `CodeLlama-34b`, `CodeLlama-7b`) for use with `RealHumanEval` to study the alignment between static benchmark performance, subjective programmer preference judgments, and programmer productivity.

of suggestions and the likelihood of copying code from chat responses, are aligned with human performance metrics. While these preference metrics are readily available in real deployments of LLM systems compared to task completion time and thus can be attractive proxy metrics (Ziegler et al., 2022), we find that they are only correlated with programmer perceptions of LLM helpfulness but not necessarily with actual programmer performance. The dissimilar findings between benchmarking and human preference metrics highlight the importance of careful evaluation to disentangle which metrics are indicative of downstream performance.

In summary, our contributions are as follows:

1. An open-source platform `RealHumanEval` to encourage more human-centric evaluations of code LLMs

2. An evaluation of 7 code LLMs of varying performance using `RealHumanEval` to provide insights into the alignment and discrepancies between benchmark performance and human preferences with downstream human performance. Our findings emphasize the importance of studying how programmers interact with code LLMs through user studies to identify nuances in programmer-LLM interactions.

3. We release the dataset of interactions collected from this study to guide the development of better coding assistants.

## 2 Related Work

Table 1: A comparison of our study against prior studies understanding programmer-LLM interactions in terms of the number of participants, models, types of LLM interaction, and tasks. Note that Cui et al. (2024) was a field experiment and thus not a controlled user study with a fixed number of tasks.

| Study | # participants | # models | Autocomplete? | Chat? | # tasks |
|---|---|---|---|---|---|
| Vaithilingam et al. (Vaithilingam et al., 2022) | 24 | 1 | ✓ | ✗ | 3 |
| Peng et al. (Peng et al., 2023) | 95 | 1 | ✓ | ✗ | 1 |
| Barke et al. (Barke et al., 2023) | 20 | 1 | ✓ | ✗ | 4 |
| Prather et al. (Prather et al., 2023) | 19 | 1 | ✓ | ✗ | 1 |
| Mozannar et al. (Mozannar et al., 2024) | 21 | 1 | ✓ | ✗ | 8 |
| Vasconcelos et al. (Vasconcelos et al., 2023) | 30 | 1 | ✓ | ✗ | 3 |
| Cui et al. (Cui et al., 2024) | 1974 | 1 | ✓ | ✗ | * |
| Ross et al. (Ross et al., 2023) | 42 | 1 | ✗ | ✓ | 4 |
| Chopra et al. (Chopra et al., 2023) | 14 | 1 | ✗ | ✓ | 4 |
| Gu et al. (Gu et al., 2024) | 22 | 1 | ✗ | ✓ | 10 |
| Kazemitabaar et al. (Kazemitabaar et al., 2023) | 69 | 1 | ✗ | ✓ | 45 |
| Nam et al. (Nam et al., 2024) | 32 | 1 | ✗ | ✓ | 2 |
| Ours | 243 | 7 | ✓ | ✓ | 17 |

*Coding Benchmarks.* Benchmarks are essential for tracking the progress of LLMs, and coding benchmarks are a key piece (Achiam et al., 2023; Laskar et al., 2023; Zan et al., 2023; Hou et al., 2023). Moreover, the coding ability of an LLM can be informative of its reasoning abilities (Madaan et al., 2022); thus, performance on coding benchmark is of broader interest. While HumanEval (Chen et al., 2021) and MBPP (Austin et al., 2021) are the most commonly used coding benchmarks, many extensions and further benchmarks have been proposed (Lu et al., 2021; Nijkamp et al., 2023; Zhu et al., 2022; Liu et al., 2023; Jimenez et al., 2023; Khan et al., 2023; Yang et al., 2024; Yan et al., 2023), we highlight a few: EvalPlus extends HumanEval's test cases (Liu et al., 2023), MultiPL-E (Cassano et al., 2023) to other languages, ReCode with robustness checks (Wang et al., 2023), HUMANEVALPACK (Muennighoff et al., 2023) with code repair and explanation tasks, and buggy-HumanEval (Dinh et al., 2023) with bugs in the reference code. Relatedly, the DS-1000 (Lai et al., 2023) benchmark evaluates models' abilities on data science problems that require using external libraries. More involved evaluations include the multi-turn program evaluation benchmark (Nijkamp et al., 2023) and SWE-bench (Jimenez et al., 2023), which requires the LLM to resolve GitHub issues. While existing benchmarks evaluate a diverse set of LLM behaviors across models, these benchmarks do not, however, include a programmer-in-the-loop, as would be the case in the increasingly common use case of programming with AI assistance. Our evaluation complements this existing line of work by conducting a user study, where programmers put the utility of these models to the test as assistants, rather than independent code generators.

*Preference Metrics.* Instead of relying solely on coding benchmarks' `pass@k` metrics, which consider only the functional correctness of LLM-generated code, recent work has advocated for incorporating human preferences, which may better reflect how LLM code could be useful to a programmer without necessarily being functionally correct (Dibia et al., 2023). Preferences are generally collected after a single turn (e.g., after a single LLM response or suggestion) and thus can be collected at scale (Bird et al., 2022; Chiang et al., 2024) or even simulated with LLMs (Dubois et al., 2023; Zheng et al., 2023). Given that preferences are only a form of intermediate feedback, in this study, we evaluate whether human preferences provide a signal for downstream productivity gains when coding with LLMs.

*Programmer-LLM Interaction.* Prior work conducting user studies where programmers code with LLM assistance has primarily focused on two forms of LLM support, autocomplete suggestions (Vaithilingam et al., 2022; Peng et al., 2023; Barke et al., 2023; Prather et al., 2023; Mozannar et al., 2024; Vasconcelos et al., 2023; Cui et al., 2024) and chat dialogue (Ross et al., 2023; Chopra et al., 2023; Kazemitabaar et al., 2023; Gu et al., 2024; Nam et al., 2024). While these studies have made progress in understanding programmer-LLM interactions, all studies only consider one LLM—often Copilot or ChatGPT—and one

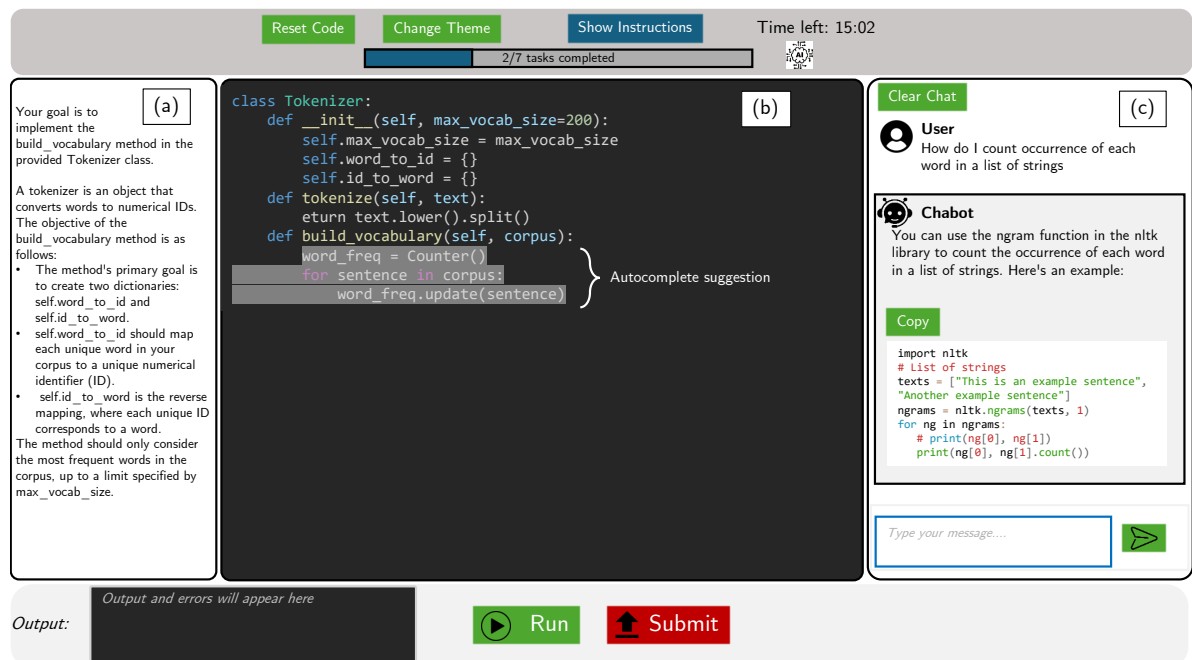

Figure 2: We introduce `RealHumanEval`, an online evaluation platform for LLM-assisted coding. The platform consists of (a) a customizable task description, (b) the code editor which shows autocomplete suggestions in grey, and (c) the chat assistant. Above the editor, users can check their task progress and the amount of time left, reset the editor, change the editor theme, and view study instructions. Below the editor, they can run and submit their code.

form of LLM support—either autocomplete or chat, making it difficult to compare outcomes and metrics *across models* and *across forms of support*. In Table 1, we compare the aspects of our study with prior works that have conducted user studies where programmers code with LLM support. To our knowledge, ours is the first study to consider models of varying performance capabilities and multiple forms of support. Additionally, we note that the majority of studies have similar participant profiles as ours (i.e., students with some programming experience and industry professions), though a few focus exclusively on novice programmers (Kazemitabaar et al., 2023; Prather et al., 2023). Finally, multiple studies have limited scope in terms of the number and types of coding tasks that are considered (e.g., focusing on one minesweeper game (Prather et al., 2023) or simple plotting tasks (Ross et al., 2023)), which differ from the breadth of tasks that have been evaluated in benchmarks and are We contribute a web platform `RealHumanEval` to enable ease of human-centric evaluation of more models and forms of support. Beyond applications of coding assistance, our study contributes to the broader literature studying human interactions with LLMs (Lee et al., 2023; Collins et al., 2023; Lee et al., 2022; Dang et al., 2022; Jakesch et al., 2023; Köpf et al., 2023; Jo et al., 2023; Brynjolfsson et al., 2023).

## 3    RealHumanEval

We introduce `RealHumanEval`, a web-based platform to conduct human-centric evaluation of LLMs for programming through the workflow shown in Figure 1. We created `RealHumanEval` to facilitate large-scale studies of programmers coding with LLMs, eliminating the need for participants to perform any additional installation of a bespoke IDE or study-specific extension or to have access to special hardware to serve study-specific models.

**Interface.** As shown in Figure 2, `RealHumanEval` incorporates many basic features of common code editors and the functionality of programming interview sites such as LeetCode. Given a coding task that consists of

a natural language description, partial code (e.g., a function signature), and unit tests that evaluate the task, `RealHumanEval` allows the programmer to write code with assistance from an LLM to complete the task. The platform has a panel that displays the natural language description of a task, as shown in Figure 2(a), alongside partial code to solve the task. Participants then write their code for the task in the code editor and can test their code with a button that checks the code against test cases and runs their code directly. The editor displays any errors, if available, and whether the code passes the unit test. Once the programmer completes the task, a new task can be loaded into the interface. For our user study, we only use a single code editor file, however, `RealHumanEval` can support multiple-file projects.

**Forms of LLM Assistance.** `RealHumanEval` supports two forms of LLM assistance: *autocomplete-based* and *chat-based*. Examples of autocomplete and chat assistants include GitHub's Copilot (Github, 2022), Replit's Ghostwriter (replit, 2023), Amazon CodeWhisperer (Amazon, 2022), and ChatGPT (OpenAI, 2022). In *autocomplete-based* assistance, the programmer writes code in an editor, and the LLM displays a code suggestion inline, which is greyed out as shown in Figure 2(b). The LLM is assumed to be able to fill in code given a suffix and prefix. A suggestion, based on the current code body in the editor, appears whenever the programmer pauses typing for more than two seconds or when the programmer requests a suggestion by pressing a hotkey. The programmer can accept the suggestion by pressing the tab key or reject it by pressing escape or continuing to type.

In *chat-based* assistance, the programmer writes code in an editor and has access to a side chat window where the programmer can ask questions and get responses from the LLM, as illustrated in Figure 2(c). The LLM is assumed to be a chat model. The programmer can copy and paste code from the LLM's responses into the editor. Currently, the interface supports any LLM invoked via an online API. Further information on the implementation of both forms of assistance is in Appendix A and Appendix C.

**Telemetry logging.** `RealHumanEval` logs all user behavior, including interactions with LLM support. For each autocomplete suggestion, we log the following tuple $\{(P_i, S_i), R_i, A_i\}_{i=1}^n$ where $(P_i, S_i)$ is the prefix and suffix of the code based on cursor position at the time of suggestion $i$, $R_i$ is the LLM suggestion, and $A_i$ is a binary variable indicating whether the suggestion was accepted. All the logs are stored in a dataset $\mathcal{D}_{ac}$. For chat-assistance, we log for each user message the following tuple $\{X_i, M_i, R_i, C_i\}_{i=1}^n$ where $X_i$ is the code at the time of message $i$, $M_i$ is the user message (including prior chat history), $R_i$ is the response from the LLM for the message, and $C_i$ is the number of times code was copied from the LLM's response. All the logs are stored in a dataset $\mathcal{D}_{chat}$. Moreover, every 15 seconds, the interface saves the entire code the user has written.

**Metrics.** From the telemetry logs, `RealHumanEval` provides multiple metrics to analyze programmer behaviors: the *number of tasks completed* (completion is measured by whether the submitted code passes a set of private test cases), *time to task success* (measured in seconds), *acceptance rate* (fraction of suggestions shown that are accepted, for autocomplete), and *number of chat code copies* (counting when user copies code from LLM response, for chat) among other metrics.

## 4 Study Design

Using `RealHumanEval`, we conducted a user study to evaluate (1) the impact of LLM assistance on programmer performance as a function of the LLM's performance on static benchmarks and (2) whether human preference metrics correlate with programmer productivity metrics.

**Overview.** For the entire duration of the study, participants are randomly assigned either to a control group, where they experienced the `no LLM` condition, or to the LLM-assisted group, where they experienced the *autocomplete* or *chat support* condition. For autocomplete-based support, the window in Figure 2(c) is hidden. For chat-based support, no autocomplete suggestions are shown in Figure 2(b). Participants are only assigned to one condition to minimize context switching, given the relatively short duration of the study. The study was conducted asynchronously using the `RealHumanEval` platform; participants were told not to use any outside resources (e.g., Google), and cannot paste any text originating outside the app into the editor. Specific instructions are in Appendix A. The first problem was a simple task (i.e., compute the sum and product of a list) for participants to familiarize themselves with the interface. Participants are given 35

| Name | Task Type | Description |
|---|---|---|
| is_bored | `Algorithmic` | Given a string of words, write a function to count the number of boredoms, where boredom is a sentence that starts with the word 'I'. |
| event_scheduler | `Algorithmic` | Given a list of events represented as a tuple (start time, end time, score), write a function that schedules the events to maximize total importance score. |
| order_by_points | `Algorithmic` | Write a function which sorts the given list of integers in ascending order according to the sum of their digits. |
| encode_message | `Algorithmic` | Write a function that encodes a message by swapping the case of all letters and replaces all vowels in the message with the letter that appears 2 places ahead of that vowel in the English alphabet. |
| triple_sum_zero | `Algorithmic` | Given a list of integers as an input, write a function that determines if there are three distinct elements in the list that sum to zero. |
| even_odd_count | `Algorithmic` | Given a number, write a function that computes both the number of even and odd digits. |
| sum_product | `Algorithmic` | Given a list of numbers, write a function that returns a tuple consisting of a sum and a product of all the integers. |
| is_multiply_prime | `Algorithmic` | Given a number, write a function that determines if the number is the multiplication of 3 prime numbers. |
| count_nums | `Algorithmic` | Given an array of integers, write a function that returns the number of elements which has a sum of digits > 0. |
| table_transform_named | `Data manipulation` | Given a dataframe with features (age, color, dates, height), write a function to transform it exactly to the following output dataframe. |
| table_transform_unnamed (2) | `Data manipulation` | Given a dataframe with unnamed features (col1, col2, col3, col4), write a function to transform it exactly to the following output dataframe. |
| t_test | `Data manipulation` | Given two arrays of numbers, write a function that compares the means of two populations. |
| retriever | `Edit & Augment` | Fill out this class which retrieves similar vectors from a collection of vectors, which supports calculating distances from a query vector, retrieving the most similar vectors to a query, and generating a similarity matrix for multiple queries. |
| login_authenticator | `Edit & Augment` | Fill out this class which manages user authentication, which supports password hashing, adding users, removing users, and changing passwords with verification. |
| calculator | `Edit & Augment` | Fill out this class which is a special calculator that keeps track of the previous operations performed and fix a set of bugs with the existing implementation. |
| tokenizer | `Edit & Augment` | Implement the build_vocabulary method, which creates dictionaries mapping words to unique IDs and vice versa and considers only the most frequent words, in the provided Tokenizer class. |

Table 2: Task descriptions for 17 tasks evaluated in study which comprise algoirthmic problems, data manipulation problems, and problems that require editing and augmenting existing code. See supplementary material for full task specification and starter code.

minutes to complete as many tasks as possible. If 10 minutes pass and the participant has not completed the task, a button appears to provide the option to skip the task.

**Tasks.** We designed 17 coding tasks for the platform that can be categorized into three categories: (a) *algorithmic problems* from HumanEval (e.g., solve interview-style coding), (b) *data manipulation problems* (e.g., wrangle input dataframe into desired output), and (c) *editing and augmenting code tasks* (e.g., fill in provided code scaffold to achieve desired behavior). While the set of tasks does not evaluate all types of coding problems exhaustively, they do capture tasks of varying difficulty and solutions of varying length, as well as the use of different programming skills, leading to varying opportunities to benefit from LLM support. A short description of each task can be found in Table 2. We chose 17 tasks to build diversity across tasks while being able to collect enough samples per task. We ensured that no LLM model considered in the study, in addition to GPT-4o, could solve all tasks perfectly, so that programmers would not simply accept all LLM suggestions and that each task could be solved in under 20 minutes by an experienced programmer (validated through pilots with the authors and volunteer participants), to ensure that these were reasonable questions to consider for a user study. These 17 tasks are distributed into five sets, where each set consists of a different mix of task types in varying orders but shares the first two tasks. Each participant is randomly assigned to one of these sets. The LLMs are not aware of the task descriptions unless the programmer types

them in the editor or chat window; this is to simulate the real world where the task description represents the programmer's hidden true intent. We provide examples of the coding tasks in Appendix B and in full in the supplementary materials.

**Conditions.** For the autocomplete conditions, we chose base LLM models that naturally generate next-word predictions, whereas the "chatty" variants of the base models are employed for the chat conditions. To evaluate the effect of LLM capabilities, we selected three types of models that demonstrate clear gaps in performance on existing benchmarks (as shown in Figure 11). In total, we selected 7 LLMs for our study: 4 from the Code Llama family (Rozière et al., 2023) (`CodeLlama-7b`, `CodeLlama-7b-instruct`, `CodeLlama-34b`, `CodeLlama-34b-instruct`), along with three models from the GPT series (Brown et al., 2020) (`GPT-3.5-turbo`, `GPT-3.5-turbo-instruct` and `GPT-4o`). To avoid confusion, we refer to the autocomplete conditions by the base name of the model: `CodeLlama-7b`, `CodeLlama-34b` and `GPT-3.5` (refers to `GPT-3.5-turbo-instruct`); and the chat conditions by the base name of the model with chat: `CodeLlama-7b (chat)` (refers to `CodeLlama-7b- instruct`), `CodeLlama-34b (chat)` (refers to `CodeLlama-34b- instruct`), `GPT-3.5 (chat)` (refers to `GPT-3.5-turbo`) and `GPT-4o (chat)`. Specific choices of parameters, system prompts, and other considerations are provided in Appendix C.

**Participants.** We recruited 263 total participants from university mailing lists and social media to capture a range of coding experiences. We verified that participants were above 18 years of age, resided in the United States, and correctly completed a simple Python screening question. Out of the 263 participants, we filtered out those who did not complete any task or did not write code for a period of 15 minutes during the study to arrive at 243 final participants. Of the 243 participants, 35% identify as Female. In terms of occupation, 80% are Undergraduate or Graduate Students studying computer science, 11% work in Software Development and 7% work in AI. While a majority of our participants were students, only 35% of participants had less than 2 years of professional programming experience. We ensured that participants were roughly equally distributed across experimental conditions based on programming experience. 11% had never used any form of AI for coding while 66% of participants use AI at least once a week for coding. Participants were provided with a $15 Amazon gift card as compensation. This study was approved by institutional IRB review.

**User study metrics.** To quantify the benefits of LLM assistance on the number of tasks completed and time to task success, we report the gap between each condition where some form of LLM assistance was provided and the control no LLM condition, which we denoted as $\Delta$. For example, for time to task success, $\Delta < 0$ for LLM support indicates that participants took less time to complete tasks with the LLM. In addition to the quantitative metrics, we also ask post-study questions to obtain participants' subjective measures of their interactions with the LLM: we ask participants to rate the helpfulness of the LLM on a scale of $[1, 10]$ and to describe how the LLM support provided (if any) was helpful and how it could be improved. We also measure two preference metrics, suggestion acceptance rate and percentage of chat code copies.

## 5 Results

We report results for data collected from 243 participants split across the seven conditions; since condition assignment is random[2], each condition has around 25 to 35 participants (except for `No LLM`, which has 39 participants). Participants completed a total of 888 coding tasks (mean of 3.6 tasks per person) on average in 358 seconds (std=153 seconds), were shown 5204 autocomplete suggestions ($|\mathcal{D}_{ac}|$), with an average 11.3% acceptance rate, and received 1055 messages from the chat LLMs ($|\mathcal{D}_{chat}|$), with 35.8% of messages having at least one copy event. In the following analyses, we conduct ordinary least squares regressions with Benjamini-Hochberg correction and use a significance level of 0.05. A more in-depth analysis of both datasets and results is in Appendix D.

**Providing LLM assistance reduces the amount of time spent coding.** To measure the productivity gains of LLM assistance to programmers, we look at two metrics: the amount of time spent coding (in seconds) and the number of tasks completed. We first distill our observations for each metric by comparing performance for each model type (i.e., combining autocomplete and chat models) against the `No LLM` condi-

---

[2]The assignment is random across all conditions except the GPT-4o condition, as that was performed individually at a later date once GPT-4o was released.

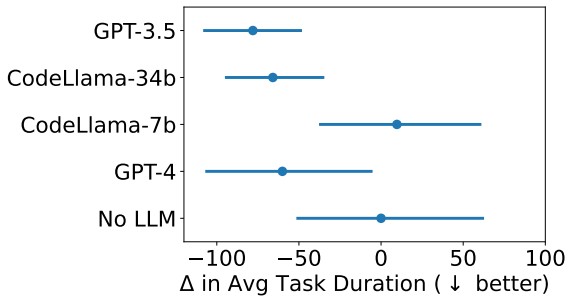

(a) Difference in task completion time (in seconds) comparing LLMs to the No LLM condition.

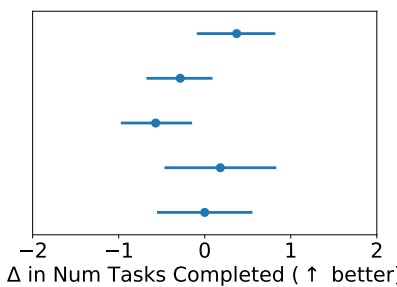

(b) Difference in number of tasks completed compared to the No LLM condition.

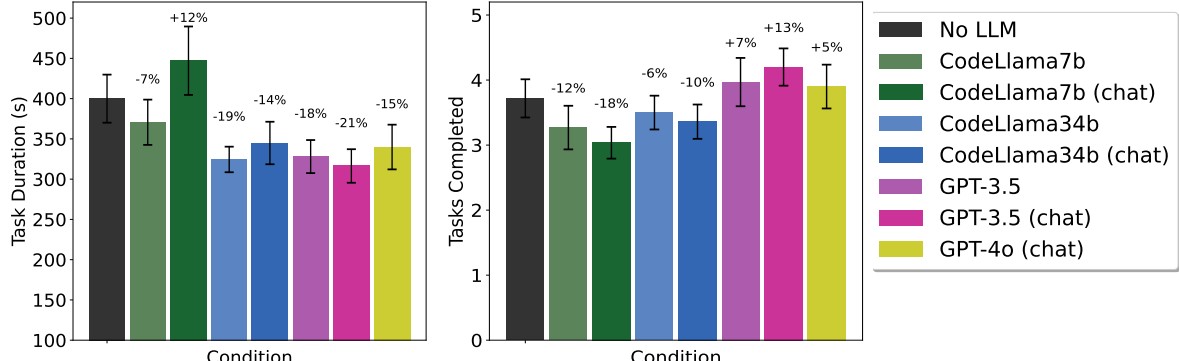

(c) Average task completion time (in seconds) by condition.

(d) Average number of tasks completed by condition.

Figure 3: We measure the effect of LLM support on user study performance on mean task duration in seconds (a,c) and number of tasks completed across model type (b,d). In (a) and (b), we compute $\Delta$, the difference between each model type—aggregating conditions corresponding to the same model type, e.g., Codellama7b and Codellama7b (chat)—and the `No LLM` condition for each metric. In (c) and (d), we break down the same metrics for each of the seven conditions and mark the percentage improvement over the `No LLM` condition. We observe that better LLM support can improve task completion time, but not necessarily increase the number of tasks completed. Error bars denote *standard errors*—the standard deviation divided by the square root of the sample size (i.e., across participants), where each participant contributes a single data point.

tion.[3] As shown in Figure 3(a), we find that compared to the `No LLM` setting where participants spent an average of 400 seconds per task, `GPT-3.5`, `CodeLlama-34b`, and `GPT-4o` models reduce the amount of time spent per task by an average of 78, 64, and 60 seconds respectively ($p = 0.04$, $p = 0.12$, and $p = 0.10$). In contrast, `CodeLlama-7b` models slightly increase the average time spent on a task by 10 seconds. However, we do not observe statistical differences across *any* of the conditions in the number of tasks completed, as shown in Figure 3(b), meaning no form of LLM support allowed programmers to solve *more* problems than they otherwise would have on their own. We hypothesize that benefits in task completion were not observed because of the short duration of the user study (35 minutes) and the amount of time it takes to complete each task, though we do observe an increase in the number of tasks attempted.

We now consider how our observations using `RealHumanEval` implicate the broader code LLM evaluation landscape, specifically the use of static benchmarks and human preference metrics.

**Are LLM performance on static benchmarks informative of user productivity with LLM assistance?** We find that improvements in model-specific evaluations on benchmarks tends to also improve human performance on both productivity metrics in the user study (i.e., `CodeLlama-7b` models led to the

---

[3]In Appendix D, we repeated the same analyses controlling for task difficulty and observed the same trends.

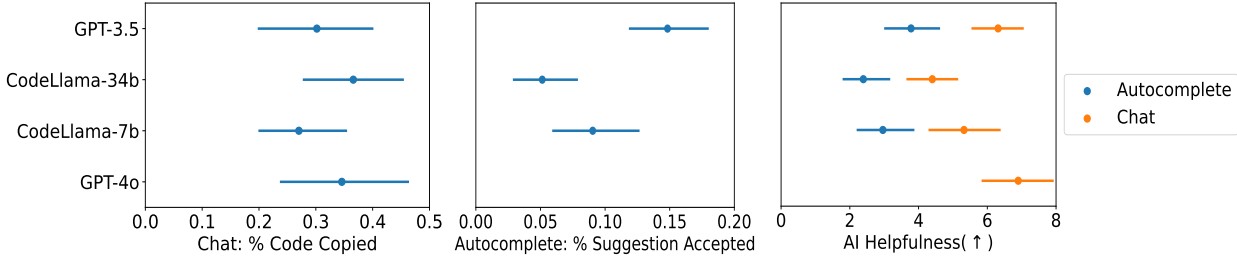

(a) Percentage of chat messages copied for chat conditions.

(b) Percentage of autocomplete suggestions accepted.

(c) Rating of LLM helpfulness across both autocomplete and chat conditions.

Figure 4: Measuring participant preferences of different models by the amount of interaction with chat (a) or autocomplete systems (b), with standard error. We find that preference judgments align with the reported helpfulness of the LLM assistant post-study (c); however, these preferences do not necessarily align with their actual task performance.

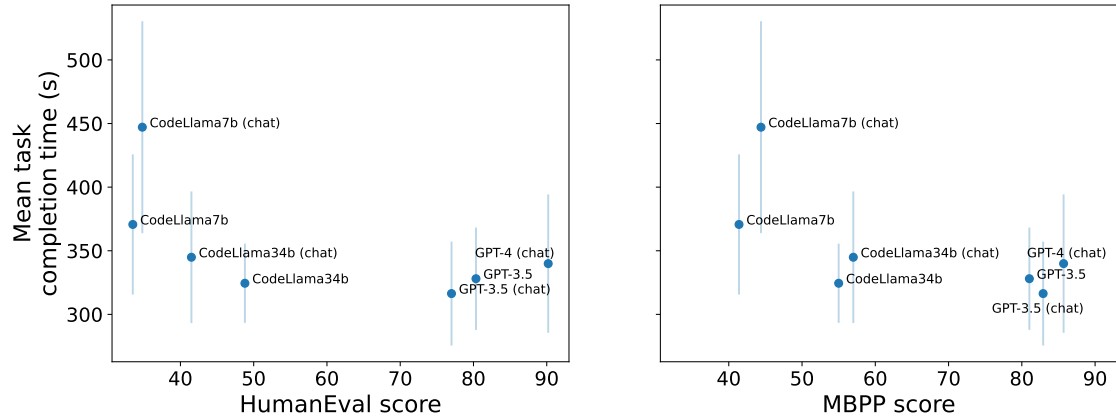

Figure 5: Average task completion time (in seconds) plotted against LLM performance on static benchmarks (HumanEval and MBPP) for each of the LLMs evaluated in RealHumanEval. Error bars denote 95% confidence intervals.

least number of tasks completed, while `GPT-3.5` models led to the most). Interestingly, this trend holds even when considering metrics with chat and autocomplete separately, in Figure 3(c-d). *However*, significant gaps in benchmark performance result in relatively indistinguishable differences in terms of human performance. In Figure 5, we plot task completion time against static benchmark score. While we do not necessarily expect performance gaps to be consistent, we find that, after a certain point, additional gains on static benchmarks may not translate to practical utility. The Pearson correlation between RealHumanEval average completion time (in seconds) and HumanEval benchmark score is -0.60 (p=0.15) and for MBPP it is -0.65 (p=0.11), which suggests a non-significant correlation, however, looking at the raw data in Figure 5 reveals the nature of the correlation. For instance, `CodeLlama-34b (chat)` is 19% better over `CodeLlama-7b (chat)` models on HumanEval, and participants are 22.8% (95% CI [2.8, 38.7]) faster on average to complete a task with 34b vs 7b. Yet, `GPT-3.5 (chat)` model outperforms `CodeLlama-34b (chat)` by 85% on HumanEval, and yet participants equipped with `GPT-3.5 (chat)` models are only 8.3% (95% CI [-11.2, 24.6]) faster than those with `CodeLlama-34b (chat)`. Surprisingly, we also find no statistically significant difference between `GPT-4o (chat)` and `GPT-3.5 (chat)` in terms of task completion time.

**Do human preferences align with productivity?** We also consider programmer preferences for the LLM assistant's suggestions on autocomplete and chat: the average suggestion acceptance rate and the average copies-per-response respectively. While `GPT-4o`, `GPT-3.5`, and `CodeLlama-34b` models reduced the amount of time spent coding over `CodeLlama-7b`, we do not find the same trends reflected in human preferences. As shown in Figure 4(a), we find that suggestions from `CodeLlama-34b` are less likely to be accepted at 5%

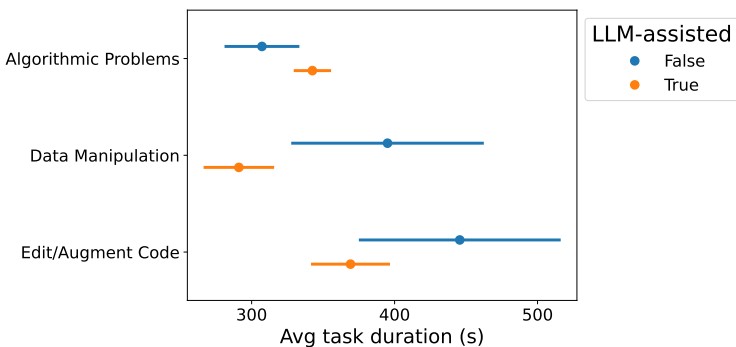

Figure 6: Average task duration with and without LLM assistance with standard error by task category.

compared to 15% and 9% for `GPT-3.5` and `CodeLlama-7b` ($p < 0.001$ and $p = 0.19$). The same ordering occurs for the percentage of chat messages copied (27% versus 35% and 29%, though not significant) in Figure 4(b). By analyzing the participants' qualitative responses, discussed in Section E, we identify potential factors that may have contributed to these preferences, including a perceived lack of context in `CodeLlama-34b` suggestions and a slight increase in latency in `CodeLlama-34b (chat)` responses. These results suggest that various external factors that might be difficult to anticipate a priori can easily affect human preferences even if they do not impact downstream productivity.

## 5.1 Additional User Study Observations

Findings on the effect of the form of LLM support and task type further illustrate the importance of evaluation with humans in the loop.

**Chat support is perceived to be more helpful than autocomplete support.** Even though autocomplete and chat variants obtained similar performance on static benchmarks and participant performance in both conditions conditioned on a model type was relatively similar, we observe that chat models are rated by participants in the post-study questions as significantly more helpful than autocomplete models ($p < 0.001$), as shown in Figure 4(c). Again, we observe that `CodeLlama-34b` models tend to be rated as less helpful (3.3 out of 10), than the other two models (4.19 and 5.09 out of 10 for `CodeLlama-7b` and `GPT-3.5`). To no surprise, `GPT-4o (chat)` is rated as the most helpful with 6.9 out of 10 followed closely by `GPT-3.5 (chat)` with 6.3 out of 10.

**The benefits of LLM assistance can vary by task type.** We also analyze the time spent on each task category, comparing when participants have access to LLM assistance versus the control condition. As shown in Figure 6, we find suggestive evidence that LLM assistance was particularly effective in reducing the time programmers needed to solve data manipulation tasks, by 26.3%, and slightly less so for problems that required editing and augmenting existing code, by 17.1%. In contrast, we found that LLMs were unhelpful on algorithmic problems, increasing the amount of time spent by 11.4%. A breakdown by individual task is in Appendix D.

**Alternative Measures of Autocomplete Suggestion Quality.** The fraction of suggestions accepted by programmers in the autocomplete conditions is a myopic measure of the interaction between the programmer and the LLM. Programmers often accept suggestions to see them with code styling and then delete them promptly or verify them at a later time (Mozannar et al., 2024). On the other hand, programmers may reject suggestions inadvertently. A measure that is less myopic than a fraction of accepted suggestions, is the persistence of an accepted suggestion in a programmer's code. In Figure 7, we track each accepted suggestion over time to see if it remains in the programmer's code. We find that suggestions from `GPT-3.5` and `CodeLlama34b` persist more frequently compared to suggestions from `CodeLlama7b` confirming our productivity assessments.

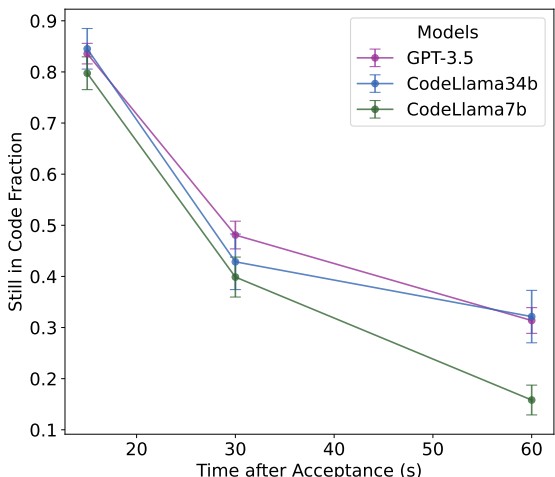

Figure 7: For each accepted suggestion across the three models for autocomplete conditions, we track after 15 seconds, 30s, and 60s whether the accepted suggestion was still found exactly in the user's code. We track the fraction of the accepted suggestions that persisted across the three time points.

## 6 Discussion

In this work, we introduce `RealHumanEval`, a human-centric evaluation platform for code LLMs, and conduct a user study using the platform to measure programmer productivity, as measured by time to complete a task and number of tasks completed, assisted by different LLMs. We now discuss how the platform can be easily used for future studies to evaluate new models and interactions as well as how the data collected from our study can be used to improve coding assistants.

**An open platform to support interactive model evaluation.** We believe `RealHumanEval` can be adopted to evaluate newly released LLM models in a more meaningful way and become a standard for evaluation. As demonstrated across several studies of AI-assisted programming, the effects of coding assistance show considerable heterogeneity by task, model, language, and user population, among other factors. RealHumanEval was built with this heterogeneity in mind, as a toolkit to enable a systematic exploration of these factors within the research community. Specifically, researchers can add their own models by providing an API endpoint, add new tasks as JSON files, and configure their own values of LLM parameters (e.g., temperature, time to trigger, etc.) through a centralized configuration file. As an open-source project, the platform is also designed to be extensible, allowing researchers to also add their own custom components to the interface with minimal friction. We will also release tutorials with the open-source platform to make it as easy as possible for others to get started.

**Improving coding assistants and leveraging study data.** We summarize participant suggestions on how coding assistants could be improved (more detail in Appendix E). Participants overwhelmingly felt that LLMs struggled to infer the appropriate *context* to provide the most useful support from the information available, highlighting the need for benchmarks that capture settings where LLMs need to infer intent from partial or fuzzy instructions. There are also opportunities to improve autocomplete and chat assistants to be better programming partners (Wu et al., 2023). For example, autocomplete systems might benefit from personalization of when participants would benefit from suggestions and dynamically adjusting the length, while chat-based systems could be improved to have better, more tailored dialogue experience and better integration with the editor. Toward these goals, we release the datasets of user interactions that can be leveraged as signals of user preferences and behavior patterns. For example, the data collected in our study presents an opportunity to build and evaluate simulation environments that mimic how programmers write code with an LLM. The data can also be used to fine-tune the models; the dataset of interactions with autocomplete models $\mathcal{D}_{ac}$ reveals which suggestions programmers accept and which they reject, which can

be used to update the LLM and generate suggestions that maximize the probability of being accepted by the programmer.

**Limitations.** Firstly, we acknowledge that a set of 17 coding tasks does not span the entire set of tasks a professional programmer might encounter in their work and may limit the generalizability of our evaluations of the 7 models. We encourage future work to leverage `RealHumanEval` to conduct further studies with a more extensive set of tasks and with more models. We included the seven models chosen to be representative of different scales of LLMs, notably we did not evaluate models from the Claude series Anthropic (2024) but we expect GPT-4o to have relatively similar performance. Our goal is not to benchmark all available LLMs, but to look at trends between human productivity and LLM performance. Furthermore, our work includes more studied models than prior work with human (Table 1). Second, the coding tasks we used are of short duration, while real-world programming tasks can take hours to months. This presents a trade-off in study design: short tasks allow us to evaluate with more participants and models in a shorter period but give us a less clear signal compared to longer-term tasks. Third, `RealHumanEval` does not fully replicate all functionality existing products such as GitHub Copilot may have so the study may underestimate exact productivity benefits. Such products are complex systems comprising more than a single LLM, where many details are hidden and thus not easily replicable. We release `RealHumanEval` to enable others to build more functionality in an open-source manner.

**Societal implications.** While our evaluations focused on productivity metrics, there are additional metrics of interest that may be important to measure when studying programmer interactions with LLM support. On the programmer side, further evaluations are needed to understand whether programmers appropriately rely on LLM support (Spiess et al., 2024) and whether LLM support leads to potential de-skilling (Bommasani et al., 2021). Further, our metrics do not consider potential safety concerns, where LLMs may generate harmful or insecure code (Pearce et al., 2022; Perry et al., 2023).

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

# Appendix

## A   User study details

### A.1   `RealHumanEval` interface screenshots

We show examples of the `RealHumanEval` web interface used in the study: autocomplete conditions (Figure 8 and Figure 9) and chat conditions (Figure 10). Note that the interface is the same as that of the autocomplete conditions for the no LLM condition, except there is no LLM to provide any inline code suggestions.

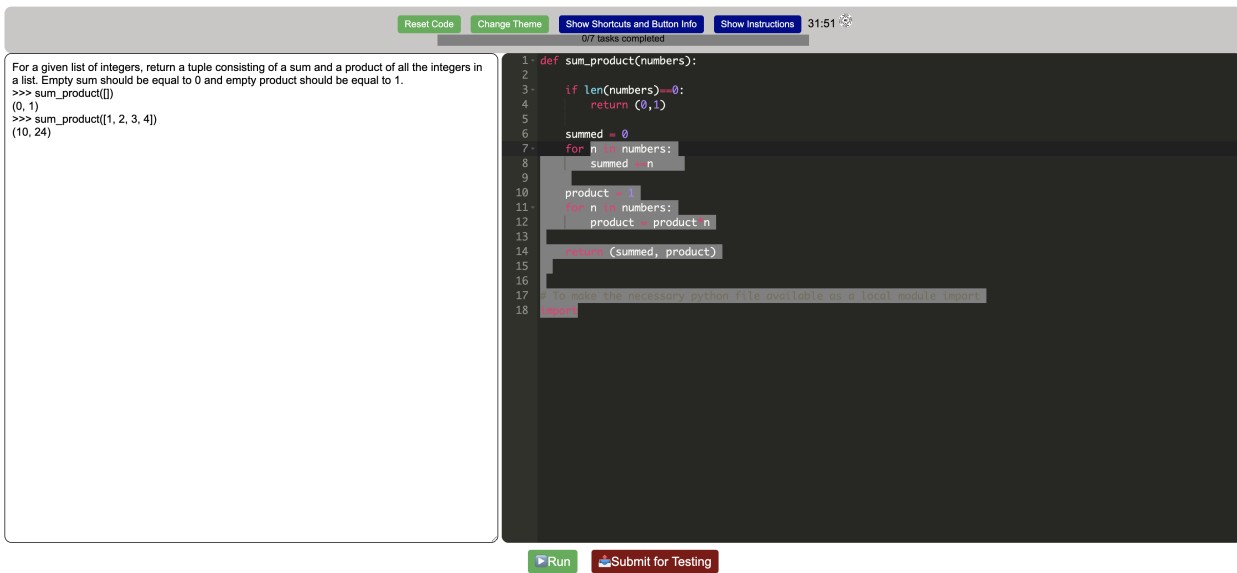

Figure 8: Screenshot of the autocomplete LLM-assistance interface in our user study.

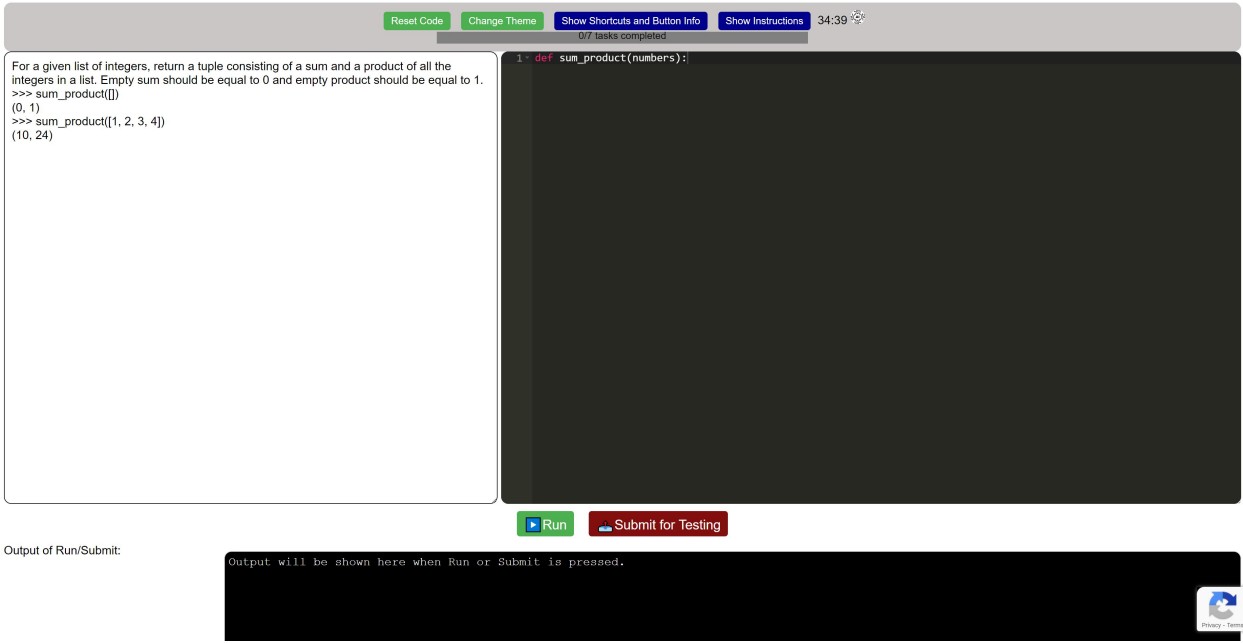

Figure 9: Another screenshot of the autocomplete LLM-assistance interface in our user study.

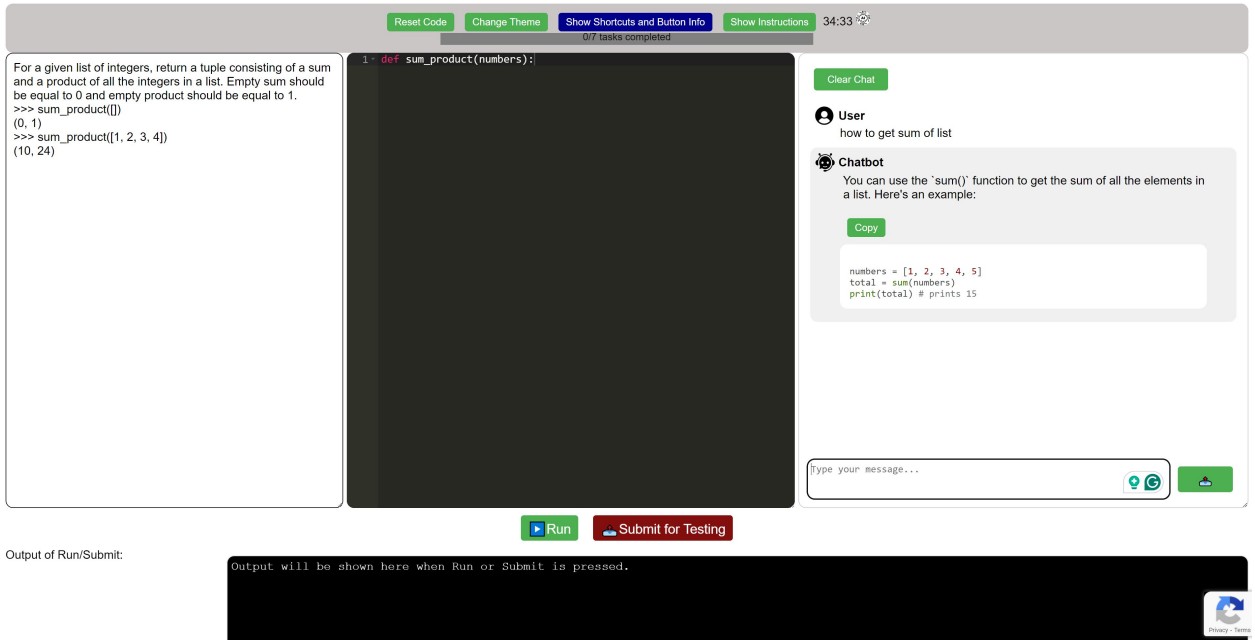

Figure 10: Screenshot of the chat LLM-assistance interface in our user study.

## A.2 User Study Instructions

Before participants enter the main interface, they are provided with the following text:

> After you fill out the information here, click the Start Experiment button to proceed.
>
> Please DO NOT refresh or press back as you may lose a fraction of your progress, if needed you can refresh while coding but you will lose your code.
>
> Your name and email will NOT be shared with anyone or used in the study.
>
> Note that there is a chance the interface may not have AI, that is not a bug.
>
> By performing this task, you consent to share your study data.

In all conditions, a pop-up is displayed with the following instruction:

Welcome to the user study! You will first complete a tutorial task to make you familiar with the study.

- You will be writing code in Python only and use only standard python libraries and only numpy and pandas.

- After the tutorial task, you will have 35 minutes total where you will try to solve as many coding tasks as possible one at a time.

- It is NOT allowed to use any outside resources to solve the coding questions (e.g., Google, Stack-Overflow, ChatGPT), your compensation is tied to effort only.

### A.2.1 Autocomplete Condition

You will write code in the interface above: a code editor equipped with an AI assistant that provides suggestions inline.

- The AI automatically provides a suggestion whenever you stop typing for more than 2 seconds.

- You can accept a suggestion by pressing the key `[TAB]` or reject a suggestion by pressing `[ESC]`.

- You can also request a suggestion at any time by pressing `[CTRL+ENTER]` (Windows) or `[CMD+ENTER]` (Mac).

- You can run your code by pressing the run button and the output will be in the output box at the bottom in grey.

- **Press the submit button to evaluate your code for correctness. You can submit your code as many times as you wish until the code is correct.**

- If you cannot solve one of the tasks in 10 minutes, a button "Skip Task", only press this button if you absolutely cannot solve the task.

Note: please be aware that the AI assistant is not perfect and may provide incorrect suggestions. Moreover, the AI may generate potentially offensive suggestions especially if prompted with language that is offensive.

### A.2.2 Chat Condition

You will write code in the interface above: a code editor equipped with an AI assistant chatbot in the right panel.

- The AI chatbot will respond to messages you send and incorporate previous messages in its response. The AI does not know what the task is or the code in the editor.

- When the AI generates code in its response, there is a COPY button that will show up above the code segment to allow you to copy.

- You can test your code by pressing the run button and the output will be in the output box at the bottom in grey.

- **Press the submit button to evaluate your code for correctness. You can submit your code as many times as you wish until the code is correct.**

- If you cannot solve one of the tasks in 10 minutes, a button "Skip Task", only press this button if you absolutely cannot solve the task.

Note: please be aware that the AI assistant is not perfect and may provide incorrect suggestions. Moreover, the AI may generate potentially offensive suggestions especially if prompted with language that is offensive.

### A.2.3 No LLM Condition

You will write code in the interface above: a code editor.

- You can run your code by pressing the run button and the output will be in the output box at the bottom in grey.

- **Press the submit button to evaluate your code for correctness. You can submit your code as many times as you wish until the code is correct.**

- If you cannot solve one of the tasks in 10 minutes, a button "Skip Task", only press this button if you absolutely cannot solve the task.

### A.2.4 Post-Study Questionnaire

- Thinking of your experience using AI tools outside of today's session, do you think that your session today reflects your typical usage of AI tools?

- How mentally demanding was the study? (1-20)

- How physically demanding was the study? (1-20)

- How hurried or rushed was the pace of the study? (1-20)

- How successful were you in accomplishing what you were asked to do? (1-20)

- How hard did you have to work to accomplish your level of performance? (1-20)

- How insecure, discouraged, irritated, stressed, and annoyed were you? (1-20)

- Overall, how useful/helpful was the AI assistant? (1-10)

- In which ways was the AI assistant helpful? What did it allow you to accomplish? (free-text)

- How could the AI suggestions be improved? (free-text)

- Additional comments (Optional): anything went wrong? any feedback? (free-text)

To ensure consistency in responses to scale-based questions, we labeled 1 with "low" and either 10 or 20 with "high" depending on the question.

### A.3 Data release considerations

We took the following measures to mitigate potential ethical concerns regarding the release of the study. First, the study protocol was approved by institutional IRB review. Second, before participating in the actual study, all participants were provided with a consent form outlining the study and the data that would be collected as part of the study (including interaction data with LLMs) and provided with the option to opt out of the study if they so choose. Finally, after data collection and prior to public data release, the authors carefully checked all participant interactions with LLMs, particularly chat dialogue, to ensure that no personally identifiable information was revealed.

## B Task Design

### B.1 Task categories

**Algorithmic coding problems:** Many coding tasks require programmers to implement algorithmic thinking and reasoning and are widely used to evaluate programmers in coding interviews. To identify algorithmic coding problems, we sample representative problems from the HumanEval dataset Chen et al. (2021). Given `gpt-3.5-turbo`'s high performance on this type of problem, we ensure that we also include problems where it fails to solve the problem on its own. We evaluated each question using test cases from the HumanEval dataset. We included the following problem ids from HumanEval: is_bored 91, is_multiply_prime 75, encode_message 93, count_nums 108, order_by_points 145, even_odd_count 155, sum_product 8, triple_sum_to_zero 40. In addition, we created a custom problem called event_scheduler. All tasks with unit tests will be released.

**Editing and augmenting existing code:** When working with existing repositories, programmers will often need to edit and build on code that may have been written by others (Sobania et al., 2023). We designed questions where participants are either provided with either code scaffold to fill in or with code body that they are asked to modify the functionality of. When designing such questions, we take care to avoid common implementations (e.g., a traditional stack and queue) that would have appeared in LLM training data. We

also constructed a set of test cases for each question. The four problem names are calculator, tokenizer, login authenticator and retriever.

For example, here is the login authenticator problem description:

> Your goal is to implement the `LoginAuthenticator` class, which will be used to authenticate users of a system. The class will include the following methods:

_hash_password (Private Method): Creates a hash of a given password. Accepts a *password* (string) and returns the hashed password using any hashing technique.

add_user Method: Adds a new user to the system with a username and a password. It checks if the username already exists, hashes the password if it does not, and stores the credentials. Returns True if successful.

remove_user Method: Removes a user from the system by deleting their username entry from `self.user_credentials` if it exists. Returns True if successful.

change_password Method: Changes a user's password after authenticating the user with their old password. If authenticated, it hashes the new password and updates `self.user_credentials`. Returns True if successful.

The programmer is given the following initial code:

```python
class LoginAuthenticator:
    def __init__(self):
        # DO NOT CHANGE
        self.user_credentials = {}  # dictionary for username: hashed_password

    def _hash_password(self, password):
        # WRITE CODE HERE
        return

    def add_user(self, username, password):
        # WRITE CODE HERE
        return

    def authenticate_user(self, username, password):
        # DO NOT CHANGE
        #Checks if the given username and password are valid
        if username not in self.user_credentials:
            return False
        return self.user_credentials[username] == self._hash_password(password)

    def remove_user(self, username):
        # WRITE CODE HERE
        return

    def change_password(self, username, old_password, new_password):
        # WRITE CODE HERE
        return
```

**Data science tasks:** Given the increased usage of data in many engineering disciplines, programmers are often involved in data science problems. We design data science problems inspired by the DS-1000 dataset Lai et al. (2023), where participants need to perform *multiple* data manipulation and wrangling operations and return a resulting Pandas dataframe. To ensure that an LLM cannot achieve perfect accuracy on its own, we only show an example of the input and target dataframes without providing specific instructions on each operation. The code will be evaluated based on the correctness of the dataframe in an element-wise fashion. The four problem names are table_transform_named, table_transform_unnamed1, table_transform_unnamed2 and t_test.

Here is for example the problem table_transform_unnamed1:

Given the input pandas DataFrame:

|   | col1 | col2 | col3 | col4 | col5 |
|---|------|------|------|------|------|
| 0 | 6 | 1 | 5.38817 | 3 | 2 |
| 1 | 9 | 2 | 4.19195 | 5 | 8 |
| 2 | 10 | 8 | 6.8522 | 8 | 1 |
| 3 | 6 | 7 | 2.04452 | 8 | 7 |
| 4 | 1 | 10 | 8.78117 | 10 | 10 |

Transform this DataFrame to match the following output structure, recognizing the relationship between the input and output DataFrames:

|   | col1 | col2 | col3 |
|---|------|------|------|
| 0 | 6 | 2 | 8.38817 |
| 1 | 15 | 3 | 9.19195 |
| 2 | 25 | 9 | 14.8522 |
| 3 | 31 | 8 | 10.0445 |
| 4 | 32 | 11 | 18.7812 |
| 0 | 0 | 0 | 0 |
| 0 | 0 | 0 | 0 |

Implement a function named `transform_df` that takes the input DataFrame and returns the transformed DataFrame, discovering and applying the patterns between them.

The programmer is given the following initial code:

```python
import pandas as pd
from io import StringIO

# Original dataset
data = '''
col1,col2,col3,col4,col5
6,1,5.3881673400335695,3,2
9,2,4.191945144032948,5,8
10,8,6.852195003967595,8,1
6,7,2.0445224973151745,8,7
1,10,8.781174363909454,10,10
'''

# Read the dataset into a DataFrame
df = pd.read_csv(StringIO(data))

def transform_df(df):
    # Your code here

print(transform_df(df))
```

## B.2 Task organization

We created five task sets where we fixed the first task (in addition to the tutorial sum_product task) and varied the remaining tasks randomly ensuring a split across the categories. The five sets are:

1. Task Set 1: even_odd_count, triple_sum_to_zero, table_transform_named, tokenizer, encode_message, t_test, event_scheduler.

2. Task Set 2: even_odd_count, is_bored, login_authenticator, is_multiply_prime, count_nums, table_transform_named, calculator.

3. Task Set 3: even_odd_count, count_nums, calculator, table_transform_unnamed2, login_authenticator, encode_message, is_bored.

4. Task Set 4: even_odd_count, order_by_points, retriever, triple_sum_to_zero, tokenizer, event_scheduler, encode_message.

5. Task Set 5: even_odd_count, is_multiply_prime, table_transform_unnamed1, t_test, is_bored, order_by_points, triple_sum_to_zero.

## C  LLM Details

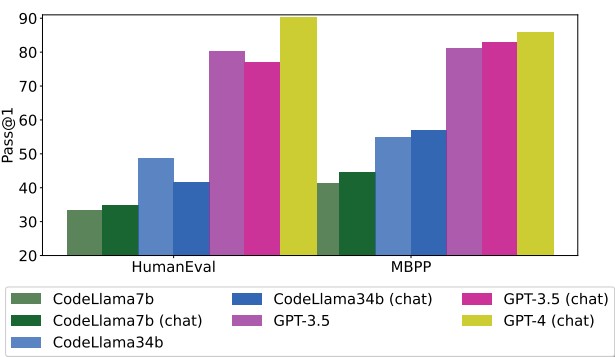

Figure 11: `Pass@1` of LLM models and their chat variants on two canonical benchmarks, HumanEval and MBPP (results from  (Rozière et al., 2023; Liu et al., 2023)), showing that `CodeLlama-7b` models perform worse than `CodeLlama-34b` models, which are less performant than `GPT-3.5` models.

We select three models of varying benchmark performance as shown in Figure 11. Here we provide links to model weights (where applicable) and any additional details.

- **CodeLlama (7b, 34b) and CodeLlama Instruct (7b, 34b).** Accessed from `https://api.together.xyz/`. Note that the base model variants are no longer available from this source. The license for the CodeLlama models is at `https://github.com/meta-llama/llama/blob/main/LICENSE`.

- **GPT-3.5-turbo.** Specific model version is `gpt-3.5-turbo-0613`. Accessed through the OpenAI API. This is a closed model and does not have an associated license.

- **GPT-3.5-turbo-instruct.** Accessed through the OpenAI API. This is a closed model and does not have an associated license.

- **GPT-4o.** Accessed through the OpenAI API. This is a closed model and does not have an associated license.

**LLM parameters.**  For all LLMs, we used a temperature setting of 1 to ensure varied responses. For autocomplete LLMs, we needed a way to set the the length of the suggestions with a fixed number since base LLMs are not trained with an EOS token and thus do not know when to stop generating code. We first looked at how current open-source Copilot systems determine the suggestion length for the autocomplete suggestions. We found that all open-source systems (such as Fauxpilot and HuggingFace's personal Copilot [4]) use a fixed-length suggestion, and each with a different length parameter. We experimented in initial study pilots with different choices and found that a token length of 64 made the suggestions more likely to be correct

---

[4]`https://huggingface.co/blog/personal-copilot`

while not being too short. However, to allow future systems to more smartly pick the suggestion length, we decided to make the suggestion length random (truncated Gaussian) on the interval [10,120] with mean 64 so that we can learn from this data in an unbiased manner. If we were to use model confidence, we would have to use an arbitrary threshold to know when to stop generation, which may be problematic in unforeseen ways. Equipped with our study data and interface, future work can pick this confidence threshold in a more sound manner by trying to maximize acceptance rate. Since the design of RealHumanEval was modular, it should be easy to plug in future mechanisms. For the chat LLMs, we set the max_token parameter to 512 tokens constant.

**Why we did not select other model candidates.** Of the CodeLlama models available to use at the time of the study, we omitted CodeLlama-13b. We did not select CodeLlama-13b as its performance on HumanEval is very similar to the 7b variant. Additionally, CodeLlama-70b and Claude-3.5-sonnet had not been released when we conducted the study.

### C.1 Prompts used

We used the following system prompt for all chat-based LLMs:

```
You are an expert Python programmer, be helpful to the user and return code
only in Python.
```

For autocomplete-based LLMs, the first line of the prompt is always the following:

```
# file is main.py, ONLY CODE IN PYTHON IN THIS FILE
```

These prompts help to ensure that LLM responds in Python.

## D Additional Results

### D.1 Pre-registration

We pre-registered our study design prior to data collection but not the analysis plan `https://aspredicted.org/blind.php?x=K3P_K1J`. We deviated from the initial plan by adding a condition with GPT-4o as a post-hoc exploration when GPT-4o was released. Due to the limit on the number of participants who completed the task within the timeframe of the study, we thus ended up with fewer participants in the final dataset than we originally anticipated being able to collect (i.e., 30 per condition instead of 50 per condition). As a result, we opted to pool together data from the same model class to study both hypotheses. All other additional analyses in this work are exploratory and were not pre-registered.

### D.2 Dataset Analysis

We post-processed both datasets to ensure they did not reveal any identifying information about participants or contain harmful language.

**Autocomplete dataset.** Recall that users had the option to request suggestions via hotkey or were provided the suggestion after some time. As shown in Figure 12, participants are much more likely to accept suggestions if they request them. Interestingly, `CodeLlama-34b` suggestions seemed to be more preferred than `CodeLlama-7b` when requested.

**Chat dataset.** We analyze the 1055 chat messages participants sent across the three conditions, as shown in Figure 13. On average 2.7 messages were sent per task with a length of 104.8 characters. We note that there is a particularly long tail in terms of words appearing in chat messages because many questions contained implementation-specific variables. In accordance with our findings that LLMs were most useful for data manipulation tasks, we also find that participants engaged with LLM support the most for those tasks.

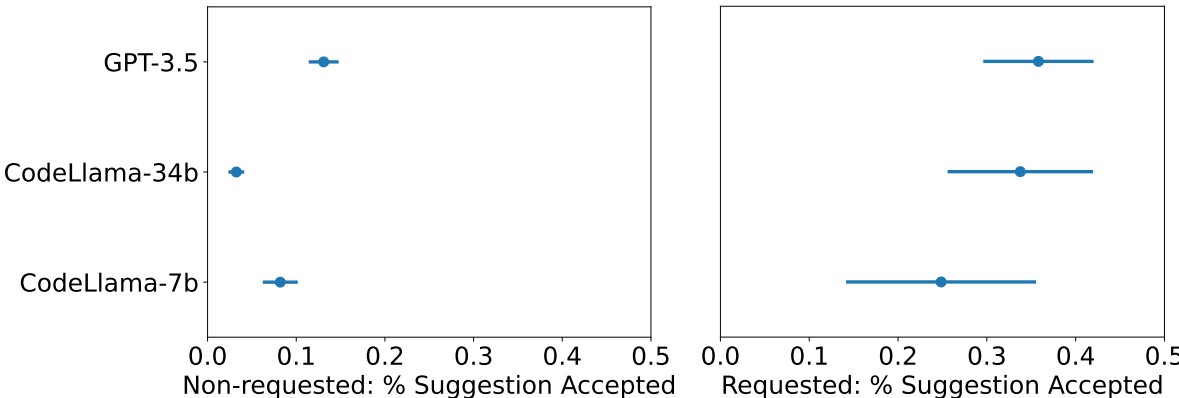

Figure 12: Comparing the acceptance rate for when participants requested suggestions with when they were automatically provided with suggestions by the autocomplete system.

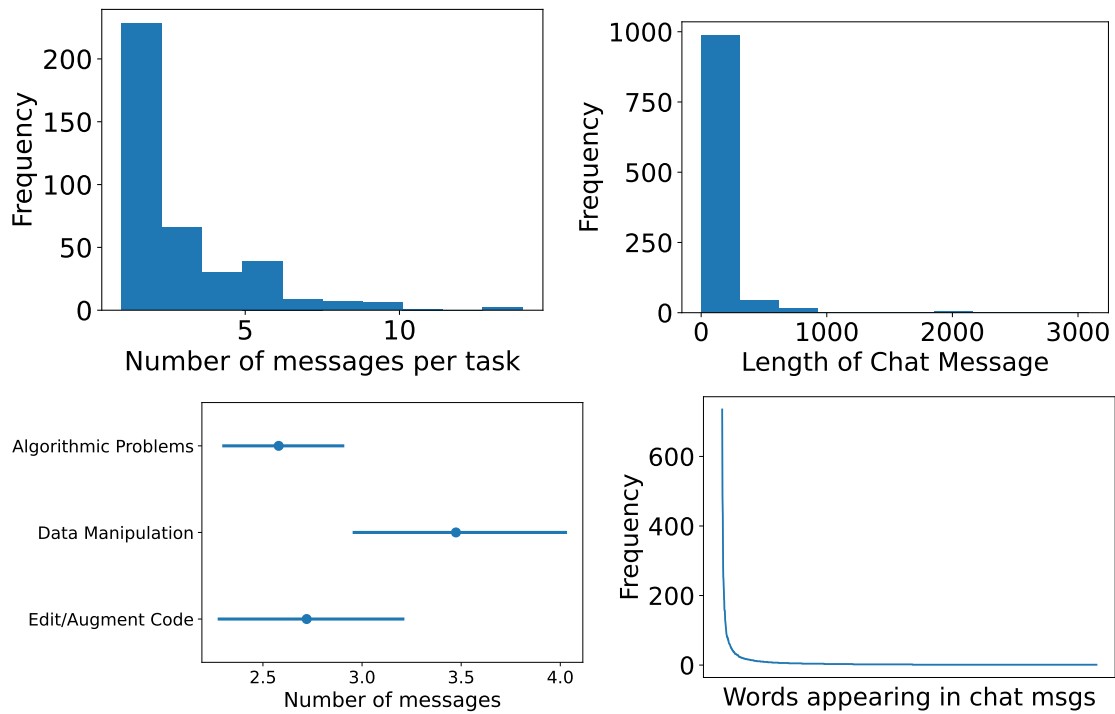

Figure 13: Analysis of the number of messages sent per task (top left), the length of chat messages (top right), the number of messages sent per task category (lower left), and the frequency of words appearing in chat messages (lower right).

### D.3 Accounting for task difficulty

To facilitate comparisons between different sets of tasks, which may have varying difficulty, the value of each metric is z-scored within the task set:

$$M_{i,t}^z = \frac{M_{i,t} - \mu_{M,t}}{\sigma_{M,t}}$$

where $M_{i,t}^z$ is the value of metric $M$ achieved by participant $i$, z-scored within task set $t$; $\mu_{M,t}$ and $\sigma_{M,t}$ are the mean and standard deviation of metric $M$ for task set $t$, across all participants. We rerun our analysis for performance metrics and present results in Figure 14.

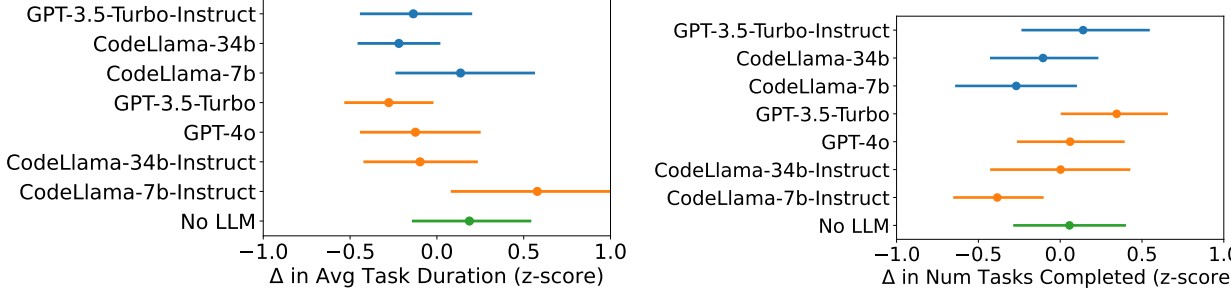

Figure 14: Performance results across models, z-scored to account for potential variation in task difficulty across sets.

### D.4 Task completion time

In Figure 3, we find the most significant differences between models in terms of task completion time. We further analyze task completion time across multiple axes.

**By task type.** When comparing when participants have access to LLM assistance versus the control condition, as shown in Figure 6, we find suggestive evidence that LLM assistance was particularly effective in reducing the time programmers needed to solve data manipulation tasks and problems that required editing and augmenting existing code, but not for algorithmic problems. We also analyze whether participants benefited from LLM assistance on an individual task level in Figure 15. We observe that trends for individual tasks within a category are similar, indicating the importance of understanding how programmers interact with LLMs for different *types* of tasks.

**Verifying outlier behavior.** We plot a histogram of task completion times in Figure 16 to verify that across participants, there were not a significant number of outliers. We also performed a similar check by plotting across conditions in Figure 18 to ensure that there was not differing behavior across participants (e.g., no bimodal behavior within a given condition).

### D.5 Code Quality Metrics

**Code Comments.** Code written with the assistance of the LLM will inherit some of the characteristics of the writing style of the LLM. One instance of that is comments in the code written. We investigate the number of comments written by participants for the different types of interaction with the LLM: autocomplete, chat, or no LLM. We count how many additional comments are in the code participants write compared to the number of comments in the provided code participants complete. Participants in the autocomplete conditions wrote $0.85 \pm 0.1$ additional comments, in the chat condition wrote $0.59 \pm 0.08$ comments and those in the No LLM condition wrote $0.41 \pm 0.13$ comments. Participants writing code with autocomplete LLM write twice as many comments as those without an LLM ($p = 3e-6$). There are two possible explanations for this increase: first, programmers usually prompt the LLM with inline comments to get a suggestion they desire, and second, we often observe that code generated by LLMs is often heavily commented. This indicates that we can potentially differentiate code written by programmers with LLM assistance by the number of comments in the code.

### D.6 TLX Results

We measure cognitive load via a series of questions from the NASA Task Load Index (TLX) Hart (2006), summarized in Table 3.

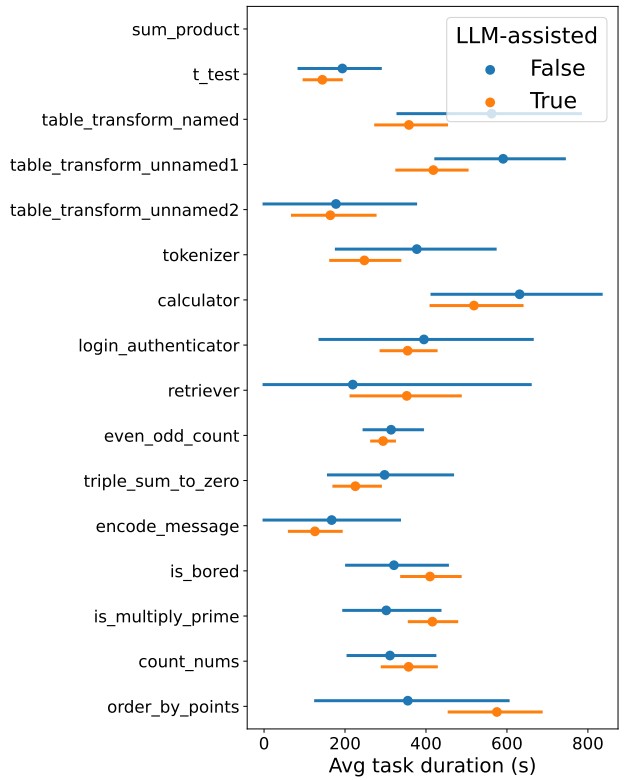

Figure 15: Time to task completion with and without LLM assistance, reported by task and grouped by task category, with standard error.

| Model | Frustration | Performance | Temporal Demand | Physical Demand | Effort | Mental Demand |
|---|---|---|---|---|---|---|
| GPT-4 | 9.83 | 8.40 | 13.00 | 3.60 | 11.33 | 12.00 |
| GPT3.5 | 8.11 | 9.11 | 12.74 | 4.71 | 11.80 | 11.37 |
| CodeLlama-34b | 13.54 | 7.96 | 11.18 | 5.18 | 10.86 | 10.93 |
| CodeLlama-7b | 11.88 | 6.50 | 13.88 | 4.88 | 10.65 | 14.50 |
| GPT3.5 (chat) | 10.09 | 9.28 | 12.19 | 4.94 | 10.88 | 12.09 |
| CodeLlama-34b (chat) | 11.04 | 8.00 | 13.44 | 5.16 | 11.40 | 12.88 |
| CodeLlama-7b (chat) | 9.54 | 7.43 | 12.57 | 6.75 | 11.93 | 11.82 |
| No LLM | 9.62 | 7.56 | 13.51 | 5.95 | 11.79 | 12.10 |

Table 3: TLX scores across conditions.

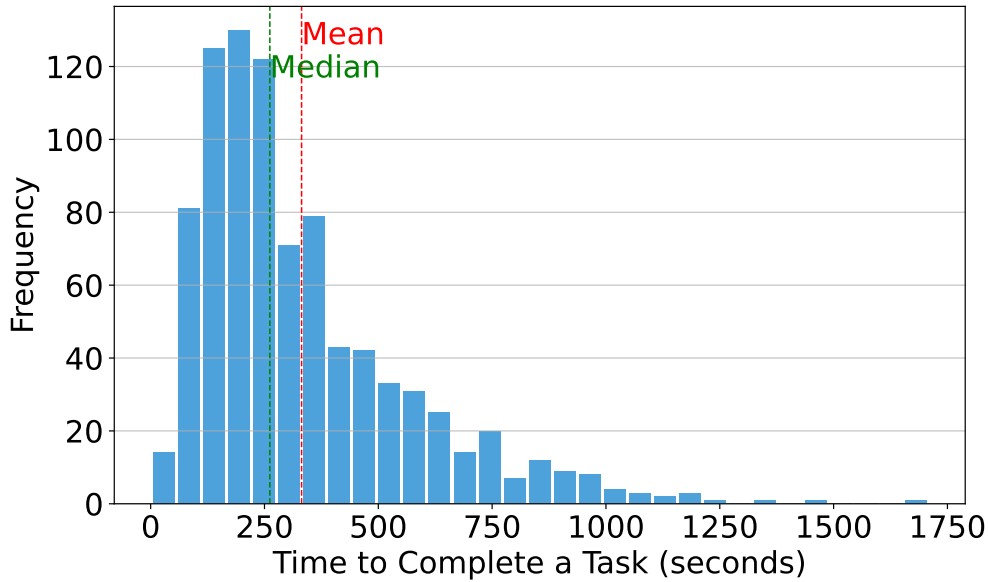

Figure 16: Histogram depicting the distribution of task completion times across all participants and conditions. The histogram is overlaid with dashed lines representing key statistical measures: the mean (red) and the median (green).

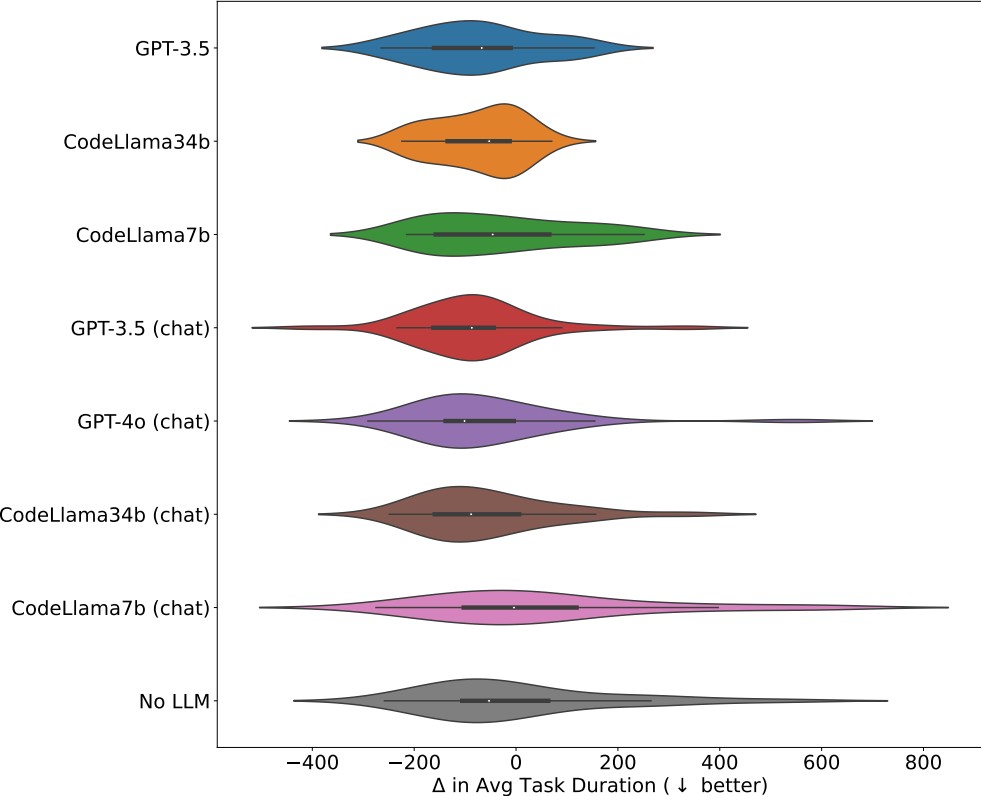

Figure 17: Violin plot of the difference in average task duration times (in seconds) between the No-LLM condition and all other conditions.

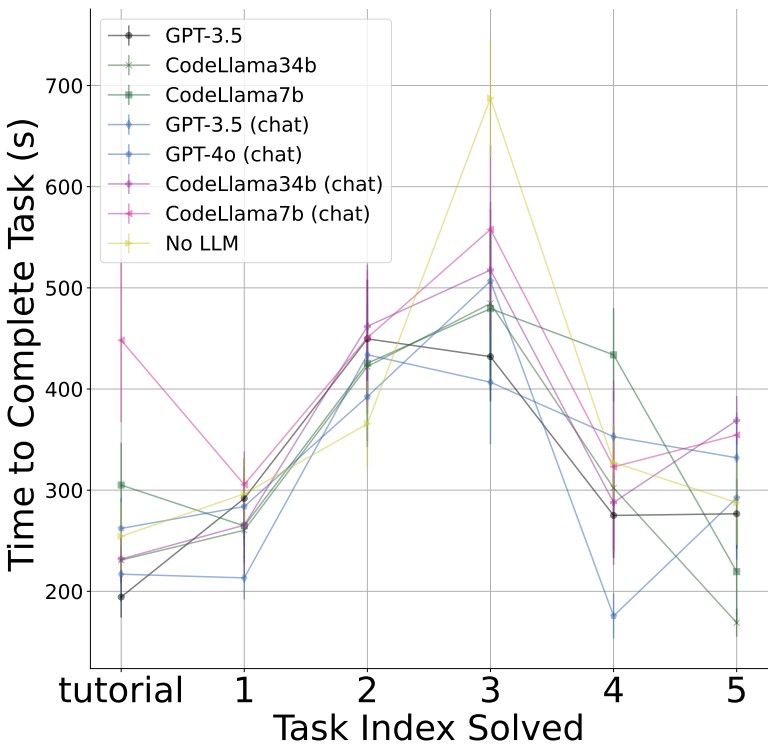

Figure 18: For each of the seven conditions, we plot the average time for participants to complete the tutorial task, the first task they solved, the second task they solved, and so on.

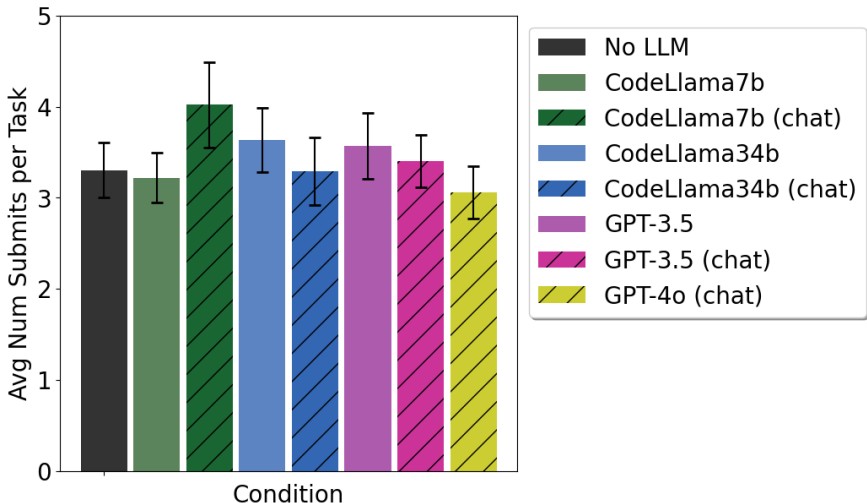

Figure 19: Average number of submissions by a participant per task across condition. We do not observe a difference in the number of attempted runs, indicating that participants did not try to brute force solutions.

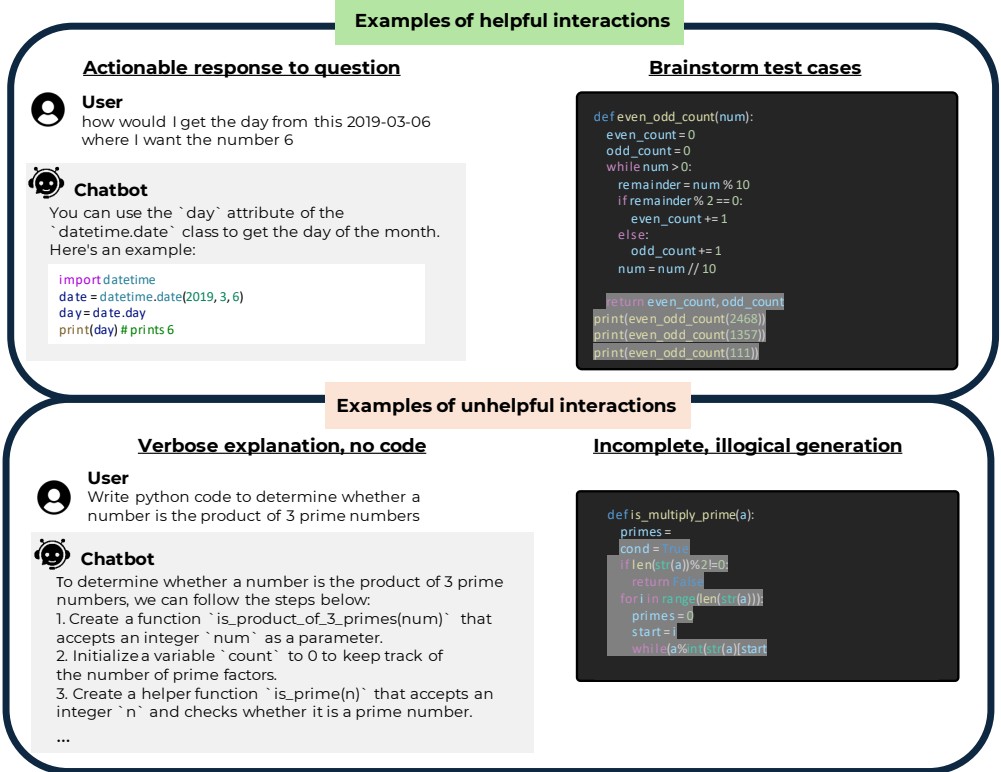

Figure 20: Examples from of helpful and unhelpful chat and autocomplete interactions from the user study. While these examples showcase how LLM assistance can improve programmer productivity (e.g., by providing actionable responses and generating test cases), they also highlight how programmer-LLM interactions can be improved. We discuss design opportunities collected from post-task participant responses in Section E and provide more examples in Appendix F.

## E   Design Opportunities

To understand the design opportunities around improving the coding assistance provided through `RealHumanEval`, we analyzed a post-study question on how coding assistants could be improved. Answers to the question were collected in free response format and were optional, though it was answered by the majority of participants. We summarize participant suggestions into general comments that could apply to both types of interactions and identify autocomplete- and chat-specific suggestions.[5]

**Both autocomplete and chat models need improved context.** A theme that spanned both types of interactions and model types was the perceived lack of context that the LLM had about the general task when providing either suggestions or chat responses (example shown in Figure 20). While one might expect that a more performant model might mitigate these concerns, we do not observe a significant decrease in mentions regarding this issue for `GPT-3.5` models compared to both `CodeLlama-7b` and `CodeLlama-34b` models. In general, it may not be obvious how to concisely specify the full "context"—recall that we intentionally considered a set-up where the LLM is unaware of task $T$ to mimic realistic constraints—but the development of new interfaces to facilitate context specification and mechanisms to prompt for additional task-specific information could improve LLM generations. Additionally, further baseline checks can be implemented to minimize concerns mentioned by participants (e.g., ensuring that the LLM responses are provided in the correct programming language, beyond prompting-based approaches implemented in our study). We note

---

[5]We omit the obvious, blanket suggestion for replacing the assistant with a better LLM, as model-only performance is one of the independent variables in our experiment and a more performant model would undoubtedly improve the assistance provided.

that issues surrounding context control have also been highlighted in prior work (Chopra et al., 2023; Barke et al., 2023).

**Autocomplete-specific suggestions.** We highlight the three most commonly mentioned avenues of improvement across all three model types. *(1) Minimize suggestion frequency:* Participants noted that the frequency of suggestions appearing in the code editor could disrupt their train of thought. To address this issue, it may be preferable to allow participants to turn off the LLM model when they are brainstorming the next steps or to modify the LLM to detect when participants may not need as frequent suggestions based on their current coding behavior. Moreover, we observe quantitatively that participants are between $3 - 10\times$ more likely to accept an assistant's suggestion *if* they requested it, as shown in Figure 12. *(2) Dynamic suggestion length:* A common issue with autocomplete interactions noted by participants was the presence of "incomplete variable definitions or function implementations" and "fragmented code" (e.g., Figure 21 (left)). As this behavior is a product of the fixed length of LLM generations, autocomplete assistants can be improved by ensuring the suggestion is complete before terminating generation. *(3) More concise suggestions:* Finally, participants also noted that code completions could be more concise, as "it was overwhelming" and "large chunks of code... start deviating from the task question" (e.g., Figure 21 (right)). It is an open question to determine the appropriate length for how much code to generate in a context-aware manner.

**Chat-specific suggestions.** There were three common suggestions shared across models. *(1) Responses should focus on code, rather than explanation:* It is well known that chat LLMs tend to generate verbose responses, which could be detrimental when used as programming assistants. An example of a lengthy response is in Figure 23. In particular, participants noted the additional time required to read large blocks of texts and suggested to "get rid of all explanations and stick to code only, unless the user specifies they want explanations." Additionally, when focusing on code, participants suggested that the chat assistant could anticipate alternative implementations *(2) Improved dialogue experience:* First, instead of making assumptions about potentially ambiguous points in a programmer's question (e.g., as in Figure 22), a participant suggested that the LLM "could ask clarifying questions or provide multiple suggestions." Additionally, in particular for `CodeLlama-7b`, participants asked for better consistency across multiple chat messages (e.g., "It wasn't able to refer back to previous messages that I had sent when answering a question."). *(3) Better integration with code editor:* Currently, the burden is on the programmer to appropriately prompt the chat assistant with questions and then to integrate chat suggestions into the code body in the editor. This onus can be reduced by more readily incorporating "the code and the most recent error, if any, as well as the test case that generated it in the context for the assistant" and "autocorrect code" based on its suggestions.

**Why was `CodeLlama-34b` less preferred by users?** Based on participants' survey responses, we identify two potential reasons that might qualitatively explain why `CodeLlama-34b` was less preferred for both autocomplete and chat. For autocomplete, the lack of context was a particularly prevalent issue in responses for `CodeLlama-34b`, mentioned by 54% of responses, as compared to 32% and 28% of `CodeLlama-7b` and `GPT-3.5` responses respectively. In particular, participants noted that the generated suggestions were often irrelevant to the prior code and in the wrong programming language. We show examples of rejected suggestions that illustrate a lack of context from participants who interacted with the `CodeLlama-34b` model in Figure 24. For chat, while there were no exceptional concerns about lack of context, `CodeLlama-34b` had the most mentions of latency as a point of improvement (6 mentions as compared to only 2 and 1 mentions for `CodeLlama-7b` and `GPT-3.5` respectively). For example, one participant noted that "the responses are slow so sometimes it was faster to go off of my memory even if I wasn't sure if it would work." Indeed, we found that `CodeLlama-34b` response time (about 10 seconds) was on average twice as slow as either `CodeLlama-7b` or `GPT-3.5` (about 5 seconds). We note that this slight delay did not significantly impact any participant's performance metrics.

### E.1 Opportunities to use data

**Simulating programmer-LLM interaction.** The data collected in our study presents an opportunity to build and evaluate simulation environments that mimic how programmers write code with an LLM. Essentially, the simulator could be used to more efficiently replicate the results of `RealHumanEval` and evaluate a wider set of models. However, despite initial work on simulating programmer-LLM interaction (Mozannar et al., 2023), building a useful simulator requires significant training and validation. Our dataset

provides training data for both chat and autocomplete interactions: The dataset of interactions with the chat models $\mathcal{D}_{\text{chat}}$ allows us to simulate the queries programmers make to the chat assistant given the code they have currently written. The dataset of interactions with the autocomplete models $\mathcal{D}_{\text{ac}}$ can allow us to simulate finer-grain interactions with LLM suggestions such as verifying and editing suggestions, among other activities outlined in (Mozannar et al., 2023). To validate a proposed simulator, one should test whether it faithfully replicates the trends observed in `RealHumanEval` before it can be used as an evaluation benchmark.

**Optimizing suggestions from human feedback.** In addition to using the human feedback data to simulate the interaction, one can use it to fine-tune the models. For instance, the dataset of interactions with autocomplete models $\mathcal{D}_{\text{ac}}$ reveals which suggestions programmers accept and which they reject, which can be used to update the LLM and generate suggestions that maximize the probability of being accepted by the programmer. Moreover, the dataset also captures how accepted suggestions were edited over time, which can be used to generate suggestions that are more likely to persist in the programmer's code. Finally, an LLM that is not instruction-tuned usually requires specifying a maximum generation length parameter to stop the generation of a code suggestion. In our autocomplete implementation, we intentionally randomized the maximum suggestion length of the generated suggestion to be between the range $[10, 120]$ with a mean token length of 64. This design decision can provide yet another signal about when the LLM should stop generating code.

## F    Example user interactions

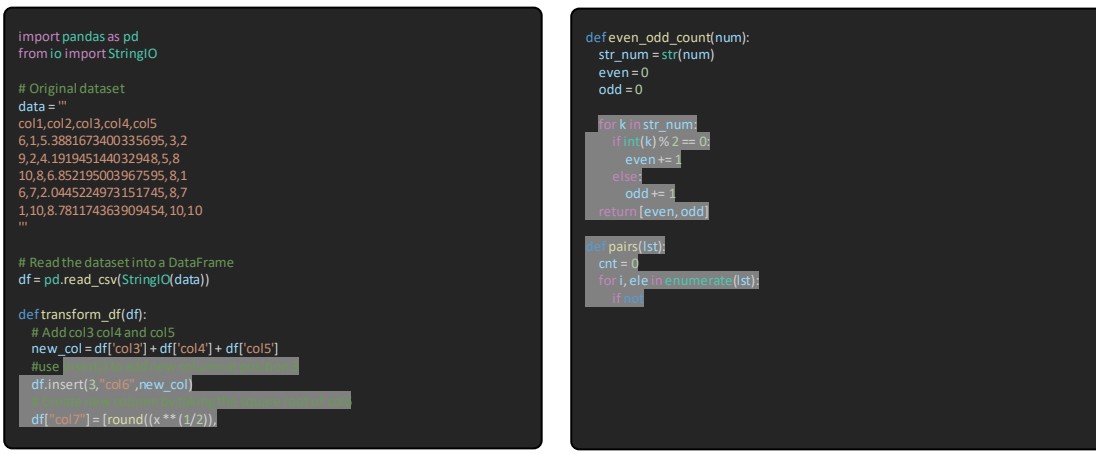

Figure 21: Examples of problematic autocomplete suggestions: incomplete suggestion (left) and starting new irrelevant function (right).

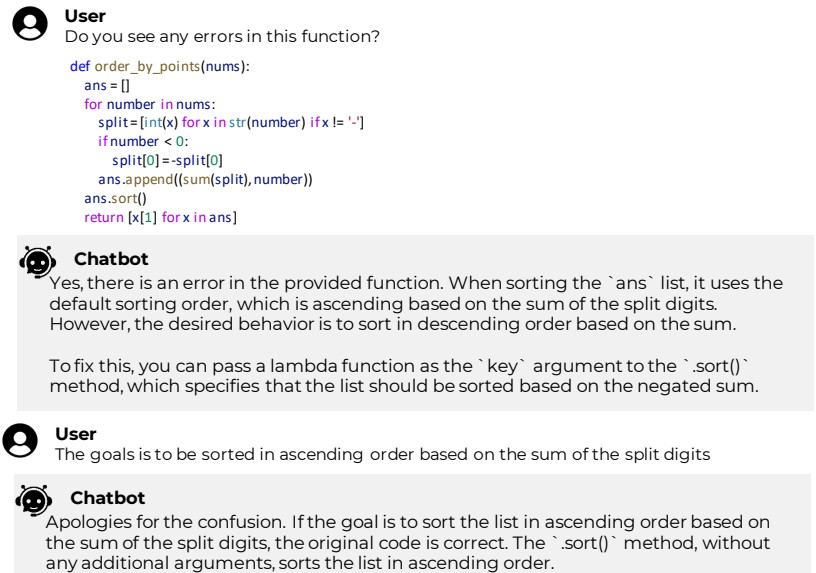

Figure 22: Example of a chat interaction where the chat assistant could have proactively asked more clarifying questions. Note that the chat agent response is shortened (by excluding code snippet) for brevity.

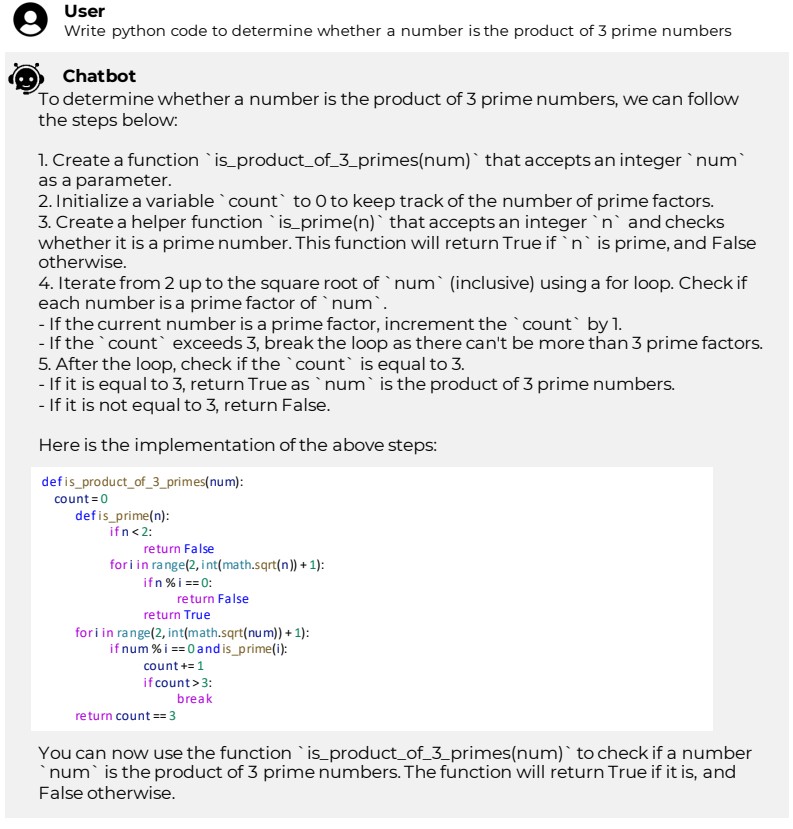

Figure 23: Example of an overly lengthy response from a chat interaction.

Figure 24: Examples of rejected suggestions from `CodeLlama-34b`, which failed to consider the context of existing code: (left) the suggested code tried to import the same packages that are already present and (right) the suggested code trails off into irrelevant, non-Python text.

