# OpenReview forum: "The RealHumanEval: Evaluating Large Language Models’ Abilities to Support Programmers"
_TMLR — Accepted by TMLR_

### Review · Reviewer_Zsge · 2024-10-24

**Summary Of Contributions:**

This paper uses real human programming procedure to evaluate the helpfulness of coding LLMs. It augment the HumanEval and other static benchmarks by measuring the human interaction metrics.

**Audience:**

Yes

**Broader Impact Concerns:**

This paper presents no detected ethical concerns.

**Claims And Evidence:**

Yes

**Requested Changes:**

The literature reviews have complete coverage, the method itself is a good supplement of current static coding LLM benchmarks, and analysis are performed. The paper looks good in current form. The scalability limitations comes from the inherent method itself which is not required to change.

**Strengths And Weaknesses:**

**Strengths**

1. The method proposed in this paper offers a new direct way to observe how LLM advancements translates to programmer productivity.

2. The paper discovered the correlations with existing LLMs Eval benchmarks, which helps to understand the quantified advancement of coding LLMs.

3. The presentation is overall good.

**Weaknesses**

1. The proposed method requires human running the evaluations, which limits the scalability

---

> ### Author Response · Authors · 2024-11-15
> **Response to Reviewer**
>
> Thank you for your constructive review. We have revised the paper and uploaded it on OpenReview, please find revised and added sections in blue text (including figures that have blue captions).

---

### Review · Reviewer_M2x2 · 2024-10-25

**Summary Of Contributions:**

This paper addresses one of the most important challenges that exists is bringing LLMs to mass-use production across many fields: evaluation.  With a focus on coding, this paper aims to bridge the gap between the understanding of performance provided by existing static benchmarks and the actual impact that a coding assistant will have on human output.  The authors produce a web interface tool that enables the measurement of how helpful a set of LLMs are for completing small coding tasks.  The interface also allows for a comparison between these models when used in the form of a chat assistant, or as an autocomplete tool.  Human study results show that stronger models according to static benchmarks are in fact more useful for helping humans complete coding tasks.  Their collected dataset is released for future public use.

**Audience:**

Yes

**Claims And Evidence:**

Yes

**Requested Changes:**

- Clarification of the scope of experiments and the conclusions that can be drawn with respect to “productivity” **(most important)**
 - More thorough explanation of how the web interface and dataset can be used by the research community in the future.

**Strengths And Weaknesses:**

**Strengths**

This paper is an ambitious effort to confront an open and important challenge, that of understanding how static benchmarking results are related to the actual usefulness of code models for human programmers.  I found it very useful to be able to make comparisons both among different models, as well as across use modes (i.e., autocomplete vs. chat).  Table 1 does a nice job of encapsulating the merits of this study.  Beyond the experimental findings, the web interface for gathering more data, as well as the resulting dataset, can stand as significant contributions as well.

**Weaknesses**

While I think that this paper does make some solid contributions, I find that it overstates the conclusions that can be drawn from their experiments, and under emphasizes the potential contribution of their web interface and dataset.

 - The use of the word “productivity” here seems like a stretch.  Productivity is something much more complex than the ability to finish 3 small coding tasks during a 35 minute sprint window.  I think most people think of productivity on a much longer timescale than this, and a study that meaningfully addressed productivity would want to consider other important questions like the effect of the tool use on programmer's skills over time.  This language should be refined or clarified, at least in some instances early in the paper, to something along the lines of “helping programmers quickly complete a series of small tasks.”  While this paper is not meant to be an in-depth discussion of the complexities of productivity, its scope should be specified in more detail, as I feel that the current broad claims around productivity lack supporting evidence.

 - I am having trouble understanding the name of the paper and dataset: RealHumanEval.  As noted in the paper, this is not the first effort to have real humans evaluate LLM code output.  Also, this work is seemingly not an extension of HumanEval (e.g., ImageNet->ImageNet-v2).

 - I do not think it is true to say that code LLMs will always be used with a programmer in the loop (e.g., top of page 4), especially on such simple tasks.  Instead, people aim to develop coding agents that can work without direct human interaction on low-level tasks.  There are many modal settings for LLM code models, and this work explores one (important) setting.

 - I think there are some issues with experiment design (e.g., participants only being assigned to one condition; short time window; small number of tasks, some of which may be overly simple and not require LLM help), but nothing disqualifying, as long as the writing correctly reflects the conclusions that can be supported.

 - I think the creation of the web interface is a strong contribution, but after reading the paper I am left wondering how the authors envision it being used in the future.  Is this an open source software tool, that someone can adopt and deploy on their own for data collection?  Or does it live on the web somewhere?  What does it take to make changes, e.g., to the set of underlying models or tasks? I think it should be clearly stated how exactly one may or may not be able to use this tool in the future.

 - I also think the dataset could represent a significant portion of the value created by this paper.  However, while it mentions the data from each interaction that is saved into the AC and Chat datasets, the paper does not explain what these datasets might be useful for.  Some indication of how these datasets may be used to produce better code models, or better evaluations for code models, would be very useful.  As it stands now, these datasets are treated more as artifacts logging your experiments than a valuable resource for future community development.

---

> ### Author Response · Authors · 2024-11-15
> **Response to Reviewer**
>
> Thank you for your constructive review. We have revised the paper and uploaded it on OpenReview, please find revised and added sections in blue text (including figures that have blue captions).
>
> To address your comments and requested changes:
>
> **Requested Changes:**
>
> - “Clarification of the scope of experiments and the conclusions that can be drawn with respect to “productivity”
>
> Thank you for the helpful suggestion to make our claims more precise! In the introduction and conclusion, we have adjusted the writing to more carefully describe how we operationalize our measure of productivity via the two metrics and after that only refer to the the metrics themselves to avoid general claims about productivity benefits. Please see the blue text for specific changes.
>
> - “More thorough explanation of how the web interface and dataset can be used by the research community in the future.”
>
> We agree the original submission did not clearly highlight these contributions and provide clear directions for future work. We have restructured the Discussion section to talk about how future work can build on both the platform and data collected from our study.
>
> **Weaknesses:**
>
> - “While this paper is not meant to be an in-depth discussion of the complexities of productivity, its scope should be specified in more detail”
>
> Thanks for the helpful suggestion to make our claims more precise! In the introduction and conclusion, we have adjusted the writing to more carefully describe how we operationalize our measure of productivity via the two metrics and after that only refer to the metrics themselves to avoid general claims about productivity benefits. Please see the blue text for specific changes.
>
> - “I am having trouble understanding the name of the paper and dataset: RealHumanEval.”
>
> The name of our platform, RealHumanEval is a play-on-words of the widely influential HumanEval benchmark: the real in "RealHumanEval" is meant to signify that it is an evaluation with **real humans**, rather than simply human-written questions. It is however not meant to be interpreted as a real evaluation in the sense that it perfectly captures all software engineering activities. We have added a footnote in the introduction to clarify this difference to avoid potential misinterpretations of our intent.
>
> - “There are many modal settings for LLM code models, and this work explores one (important) setting.”
>
> We agree and have modified the language on top of page 4 accordingly.
>
> -  “Some issues with experiment design (e.g., participants only being assigned to one condition; short time window; small number of tasks, some of which may be overly simple and not require LLM help)”.
>
> We note that “participants only being assigned to one condition” is a deliberate choice to help us quantify the benefit of a given model. However, we agree with the reviewer on the other points and have already added language on this in the Limitations section of submission.
>
> - “I think the creation of the web interface is a strong contribution, but after reading the paper I am left wondering how the authors envision it being used in the future.” “I also think the dataset could represent a significant portion of the value created by this paper.”
>
> We agree the original submission did not clearly highlight these contributions and provide clear directions for future work. We have restructured the Discussion section to talk about how future work can build on both the platform and data collected from our study.

---

> > ### Comment · Reviewer_M2x2 · 2024-11-15
> > **Updated Rating**
> >
> > Thank you to the authors for considering my review and updating the manuscript.
> >
> > My comment w.r.t. the title still stands.  I understood that you are trying to play on HumanEval, but being that the datasets bear no strong relation beyond LLMs and coding, it seems like a stretch, with the hope to gain attention by association.
> >
> > However, I feel that my most important concerns have been addressed, and updated my ratings accordingly.

---

### Review · Reviewer_tH1K · 2024-11-05

**Summary Of Contributions:**

This paper reports on a human evaluation (N=243 participants) of code LLMs, where human programmers receive tasks that to complete. Across 7 condition (one with no LLM assistance, 3 with instruction-tuned LLMs under an autocomplete feature, and 3 with chat LLMs, from CodeLlama variants to GPT-4o), participants were given time to complete as many tasks as possible, from a pool of 17 task. Generally, LLM assistance seems to help participants finish tasks faster, thus gaining in productivity. However, the productivity gains do not fully reflect differences in the static evaluation of HumanEval (large HumanEval gaps don't translate do large productivity gaps). The authors will release the RealHumanEval platform to support future human-centric evaluations of code LLMs.

**Audience:**

Yes

**Broader Impact Concerns:**

No concerns

**Claims And Evidence:**

Yes

**Requested Changes:**

* Please expand more in the main text about the actual tasks. Since the authors have space left, I'd suggest a table with 1-2 sentence descriptions of each task.
* I'd also suggest a simple correlation analysis between HumanEval performance and the productivity metrics in RealHumanEval, to accompany the corresponding finding in Section 5
* The numbering of the findings is a bit confusing, since "Providing LLM assistance reduces the amount of time spent coding." is not numbered, but is the first result; then, the next one is labeled (1). In 5.1, the numbers are dropped. Perhaps the authors can standardize these headings, as they find fit.
* [Minor] There are some occurrences of "GPT-4" in the text (as opposed to 4o): e.g., in Section 4 paragraph "Tasks", Section 5 paragraph "Providing LLM assistance ...".
* [Minor] Fig 6 caption missing a space before "seconds"

**Strengths And Weaknesses:**

# Strengths

* This kind of human-centric study is very important to understand the actual impact of LLMs.
* The RealHumanEval platform that the authors will release will be potentially useful for future studies of this kind.
* The paper is very well written and the study was rigorously executed. I can't point to any important issues in the methodology.
* The study was pre-registered (though with differences in the analysis and findings, but still better to have some pre-registration in any case!)

# Weaknesses

* I missed having a more concrete sense of the actual 17 tasks. I saw the problem IDs in the Appendix, but it'd have helped to have at least a short description of the actual tasks to be completed.
* I was also expecting an analysis correlating each LLMs performance on HumanEval (or other benchmark), vs its impact on productivity. That is discussed in the text in finding 1 ("(1) Are LLM performance on static benchmarks informative of user productivity with LLM assistance?"), but pointing at specific data points.
* One bit about the results seem puzzling: LLM assistance reduces time in each task, but doesn't generally increase the number of tasks completed. I suggest the authors clarify why this is the case. I'd suspect that the time saved in the duration of the study is generally not enough for a completely new task, due to duration of each task and the overall study. If so, this seems just a scaling issue (i.e., a longer study would see corresponding impact in both metrics), and perhaps can just clarify this (unless I missed it).

---

> ### Author Response · Authors · 2024-11-15
> **Response to Reviewer**
>
> Thank you for your constructive review. We have revised the paper and uploaded it on OpenReview, please find revised and added sections in blue text (including figures that have blue captions).
>
> To address your comments and requested changes:
>
>
> **Requested Changes:**
>
> - “Please expand more in the main text about the actual tasks. Since the authors have space left, I'd suggest a table with 1-2 sentence descriptions of each task.”
>
> Thank you for this suggestion, we added Table 2 which provides short task descriptions for each of the 17 tasks.
>
> - “I'd also suggest a simple correlation analysis between HumanEval performance and the productivity metrics in RealHumanEval, to accompany the corresponding finding in Section 5”
>
> We added Figure 5 to the paper that shows a raw plot of RealHumanEval metrics to HumanEval and MBPP performance. Additionally we computed the Pearson Correlation between RealHumanEval human performance numbers and static benchmark score and found a non-significant correlation but one needs to look at the raw data in Figure 5 to reveal the nature of the correlation.
>
> - “The numbering of the findings is a bit confusing, since "Providing LLM assistance reduces the amount of time spent coding." is not numbered, but is the first result; then, the next one is labeled (1). In 5.1, the numbers are dropped. Perhaps the authors can standardize these headings, as they find fit.”
>
> We have removed the numbering of the findings to reduce confusion, thanks for the suggestion to improve clarity!
>
> - [Minor] There are some occurrences of "GPT-4" in the text (as opposed to 4o): e.g., in Section 4 paragraph "Tasks", Section 5 paragraph "Providing LLM assistance ...".
>
> We have fixed this issue, thank you for pointing it out.
>
>
> - “[Minor] Fig 6 caption missing a space before "seconds"”
>
> We have also addressed the minor formatting issues.
>
> **Weaknesses:**
>
> - “One bit about the results seem puzzling: LLM assistance reduces time in each task, but doesn't generally increase the number of tasks completed.”
>
>  We briefly discuss this at the end of the result on “Providing LLM assistance reduces the amount of time spent coding.” Indeed, our hypothesis is exactly aligned with the reviewer. We acknowledge this as a limitation of the current study and encourage future work to explore the same metrics via a longer study.

---

### Decision · Action_Editor_gmci · 2024-12-23

**Recommendation:** Accept with minor revision

**Comment:**

The AE also resonates with one of the reviewers' comments on the benchmark name. RealHumanEval is a bit overclaiming than what the benchmark actually entails. Given that there are so many (static) benchmarks in various open-source platforms, such as HuggingFace, the AE believes that the name should entail the scope.

**Audience:**

Given the recent attention on improving language models' capabilities in various tasks, most thought that humans are better at these jobs. The human-centric benchmark and evaluation platform this paper introduces could be of the community's interest in the long run.

**Claims And Evidence:**

The key contribution of this paper is to introduce a human-centric benchmark for evaluating the capability of large-language models to support programming. The reviewers and AE believe that studies across different models and static benchmarks bring open questions to the community on what we've been improving on these models' code generation capabilities. The work also opens the framework used to conduct studies, which will be useful for future research on the same subject.